# Flotillin-mediated stabilization of unfolded proteins in bacterial membrane microdomains

Marta Ukleja[1], Lara Kricks[1], Gabriel Torrens[2,3], Ilaria Peschiera[1], Ines Rodrigues-Lopes[4], Marcin Krupka[1], Julia García-Fernández[1], Roberto Melero[5], Rosa del Campo[6], Ana Eulalio[4,7], André Mateus[3,8], María López-Bravo[1], Ana I. Rico[1], Felipe Cava[2,3] & Daniel Lopez[1] ✉

The function of many bacterial processes depends on the formation of functional membrane microdomains (FMMs), which resemble the lipid rafts of eukaryotic cells. However, the mechanism and the biological function of these membrane microdomains remain unclear. Here, we show that FMMs in the pathogen methicillin-resistant *Staphylococcus aureus* (MRSA) are dedicated to confining and stabilizing proteins unfolded due to cellular stress. The FMM scaffold protein flotillin forms a clamp-shaped oligomer that holds unfolded proteins, stabilizing them and favoring their correct folding. This process does not impose a direct energy cost on the cell and is crucial to survival of ATP-depleted bacteria, and thus to pathogenesis. Consequently, FMM disassembling causes the accumulation of unfolded proteins, which compromise MRSA viability during infection and cause penicillin re-sensitization due to PBP2a unfolding. Thus, our results indicate that FMMs mediate ATP-independent stabilization of unfolded proteins, which is essential for bacterial viability during infection.

Cellular membranes influence all cellular processes, yet we still need a better understanding of their organization. The pioneering *fluid mosaic model*, which proposed that membrane proteins and lipids diffuse freely and distribute homogenously[1], is only an approximation of our current understanding of cell membrane organization. Cell membranes are a heterogeneous mixture of lipids and proteins[2,3], some of which segregate into microdomains or lipid rafts[4]. Lipid rafts are cholesterol- and sphingolipid-enriched microdomains in eukaryotic membranes, which accumulate proteins associated with cell signaling or trafficking. Since the function of the raft-harbored

proteins requires the scaffolding activity of the raft-associated protein flotillin[5–7], many human pathologies, including neurodegenerative diseases, are associated with flotillin malfunction[8]. Due to the variable size (20–200 nm) and dynamic nature, the very existence of membrane rafts remained controversial until recent imaging and microscopy methods allowed their direct visualization[9–17]. Technical progress in the field led to the description of raft-like membrane microdomains in diverse organelles[18,19]. These microdomains require the aggregation of distinct lipids (i.e., cholesterol, sphingolipids, gangliosides, or cardiolipin), their colocalization with flotillin-like

[1]Department of Microbiology, National Centre for Biotechnology, Spanish National Research Council (CNB-CSIC), Madrid 28049, Spain. [2]Department of Molecular Biology, Umeå University, Umeå SE-901 87, Sweden. [3]The Laboratory for Molecular Infection Medicine Sweden (MIMS). Umeå Center for Microbial Research (UCMR). Science for Life Laboratory (SciLifeLab), Umeå SE-901 87, Sweden. [4]Center for Neuroscience and Cell Biology (CNC), University of Coimbra, 3004-504 Coimbra, Portugal. [5]Department of Structural Biology, National Centre for Biotechnology, Spanish National Research Council (CNB-CSIC), Madrid 28049, Spain. [6]Instituto Ramón y Cajal de Investigación Sanitaria (IRYCIS), Ramón y Cajal Hospital, 28034 Madrid, Spain. [7]Department of Life Sciences, Center for Bacterial Resistance Biology, Imperial College London, London SW7 2AZ, United Kingdom. [8]Department of Chemistry, Umeå University, Umeå SE-901 87, Sweden. ✉e-mail: dlopez@cnb.csic.es

proteins[20], and the accumulation of a pool of proteins distinctive to each organelle[18,19]. The alteration of flotillin affects the function of the organelles and induces cellular malfunction programs, such as apoptosis, inflammation, or autophagy[21–24]. However, the contribution of membrane microdomains to cellular functions still needs to be clarified.

The existence of membrane microdomains has traditionally been considered a fundamental step during the evolution of cellular complexity associated exclusively with eukaryotic cells. However, we and other laboratories proved this is a universal organization principle affecting both prokaryotic and eukaryotic membranes. Prokaryotic membranes organize functional membrane microdomains (FMM)[25,26], which require the aggregation of isoprenoid lipids and cardiolipin into highly hydrophobic membrane regions and their colocalization with flotillin-homolog proteins[26,27]. As seen with eukaryotic rafts, a particular pool of *client proteins* is associated with bacterial FMM. FMM perturbation using flotillin mutants compromises the activity of the FMM client proteins, leading to defective bacterial processes, such as virulence and antibiotic resistance[28–34], biofilm formation, sporulation, motility[26,27,35], membrane fluidity[36,37], heme acquisition[38] and impaired thylakoid integrity in cyanobacteria[39]. Nonetheless, the molecular function of FMM or flotillin in bacterial biology is unclear[37], as is in eukaryotes. However, bacteria are tractable organisms that provide unique models for exploring the mechanistic aspects of membrane microdomains. Specifically, the human pathogen *Staphylococcus aureus* expresses a single flotillin (*floA*), and the synthesis of isoprenoid FMM lipids is well-known[40–42]. In addition, *S. aureus* attracts much attention due to its remarkable ability to overcome antibiotic treatments[43]. Methicillin-resistant *S. aureus* (MRSA) isolates are common in hospitals and show resistance to penicillin and other antibiotics[44,45]. As there are limited resources to treat MRSA infections adequately[43], MRSA is a world-leading pathogen for death associated with antibiotic resistance[46].

The visualization of FMM in MRSA cells has been described as the accumulation of FloA in membrane microdomains enriched in staphyloxanthin (STX) and depleted in phospholipid content using a combination of electron tomography and super-resolution fluorescence microscopy[29]. Moreover, the MRSA Δ*floA* mutant shows reduced virulence and higher sensitivity to penicillins in a murine infection model[29,30], a phenotype associated with a general malfunction of the FMM-associated proteins. One FMM client protein that interacts with FloA is PBP2a[29], a low-affinity penicillin-binding protein (PBP) responsible for penicillin resistance in MRSA. Penicillins inhibit PBP and peptidoglycan synthesis[47], but PBP2a allows MRSA to grow and divide in the presence of these antibiotics[45]. The inhibition of flotillin or the FMM lipid STX in MRSA causes the inactivation of PBP2a, resulting in MRSA infections susceptible to penicillin[29]. How FMM and flotillin contribute to preserving PBP2a activity is unknown.

Here, we show that FMM are membrane compartments dedicated to accumulating unfolded proteins caused by cellular stress, to stabilize and isolate them to prevent unspecific interactions. We discovered that FloA assembles in a clamp-shaped conformation that holds the unfolded client proteins, such as PBP2a, to favor its correct translocation and folding, thus preserving MRSA antibiotic-resistant phenotype. The recruitment of unfolded proteins to FMM and flotillin contribution to protein stabilization are driven by hydrophobic interactions and do not involve a direct ATP cost to the cell. FMM are thus critical cellular structures to bacterial survival in conditions of ATP depletion, such as during infection. Targeting FMM led unfolded proteins to misfold, compromising bacterial viability and additionally causing PBP2a misfolding, leading to their re-sensitization to penicillins of multi-drug resistant MRSA clinical isolates during in vivo infections. Our work unravels the function of FMM as bacterial membrane compartments dedicated to ATP-independent protein stabilization and highlights their essential role in maintaining bacterial viability during infections.

## Results

### FloA and NfeD are essential for bacterial survival under stress conditions

Flotillin genes are present in almost all bacterial genomes as a second gene of an operon, in which the first gene codes for a membrane protein (NfeD) that is exclusive to prokaryotes[48–50] (Supplementary Fig. 1A and Supplementary Dataset 1). The biological function of NfeD is unknown but its cytosolic domain faces the cytoplasmic side of the membrane (Supplementary Fig. 1B), where is known to interact with flotillin (FloA)[51,52], suggesting that FloA and NfeD are functional partners. We generated Δ*nfeD* and Δ*floA* mutants in the MRSA clinical isolate USA300 (Supplementary Fig. 1C–E). In in vitro phenotypic assays, both mutants showed a significant reduction of colony-forming units (CFU) count compared to the wild-type strain (WT) in the presence of oxidative stress ($H_2O_2$ 5 mM) (Fig. 1A). In contrast, no differences in CFU count were detected between the complemented strains and the WT (Supplementary Fig. 1E) or non-stressed laboratory cultures (Supplementary Fig. 1F), suggesting that FMM play a role against infection-related stress.

Strains labeled with FloA-YFP or NfeD-YFP (yellow fluorescent protein) translational fusions (Supplementary Fig. 1E) showed the fluorescent signal distributed in discrete foci across the cell membrane (Fig. 1B). Both $H_2O_2$-stressed cells and untreated cells collected at stationary phase showed an increase in the number of foci compared to untreated, exponentially-growing cells. Since oxidative stress is central to the antimicrobial response of innate immune cells[53,54], we explored the role of *floA* and *nfeD* during in vitro infection of stress-induced macrophages (Fig. 1C). We used fluorescence microscopy and CFU count to monitor bacterial load at 1.5 h post-infection. The Δ*floA* and Δ*nfeD* mutants showed lower intracellular levels within infected host cells ($H_2O_2$ 1 mM) compared to WT. The relevance of FloA or NfeD in *S. aureus* virulence was tested in vivo using the invertebrate *Galleria mellonella* model (Fig. 1Di). This model shows an immune system constituted by phagocytic cells, which typically use oxidative stress to kill infecting bacteria[55]. Larvae were infected with WT, Δ*floA* or Δ*nfeD* strains ($10^6$ CFUs) and were incubated at 37 °C for 48 h upon counting for survival. Infection of larvae with WT resulted in a survival rate of ~15%. In contrast, larvae infected with Δ*floA* or Δ*nfeD* showed an 80% survival rate, pointing to a reduced virulence in Δ*floA* or Δ*nfeD* strains. Furthermore, using a mouse sepsis infection model (Fig. 1Dii–iv), mice were infected intraperitoneally with WT, Δ*floA* or Δ*nfeD* strains ($3 × 10^7$ CFUs). The infections were maintained for two days before animals were sacrificed, their organs were collected and bacterial load counted. Mice infected with Δ*floA* or Δ*nfeD* showed significantly reduced bacterial load compared to those infected with WT. In sum, FloA and NfeD are critical for bacterial survival during infection.

### FloA and NfeD accumulate and oligomerize at FMM

The function of FloA and NfeD in preserving bacterial viability during infection was addressed structurally and functionally. Structurally, Alphafold2 (AF2) prediction[56] showed FloA to have one amino terminal ($N_t$) transmembrane segment adjacent to a cytoplasmic prohibitin domain (PHB) of unknown function present in prokaryotic and eukaryotic flotillins (Fig. 2Ai and Supplementary Movie 1). The PHB domain contains a subdomain-1 of six parallel α-helices that vary over the phylogenetic tree and a conserved subdomain-2 organized in three antiparallel α-helices packed perpendicularly against a three-stranded antiparallel β-sheet[57] (Supplementary Fig. 2A). After the PHB domain, FloA presents a coiled-coil domain (CC region) linked to a low-complexity region (LCR region)[50]. AF2 prediction showed NfeD to possess a $N_t$ region of six transmembrane segments rich in aromatic

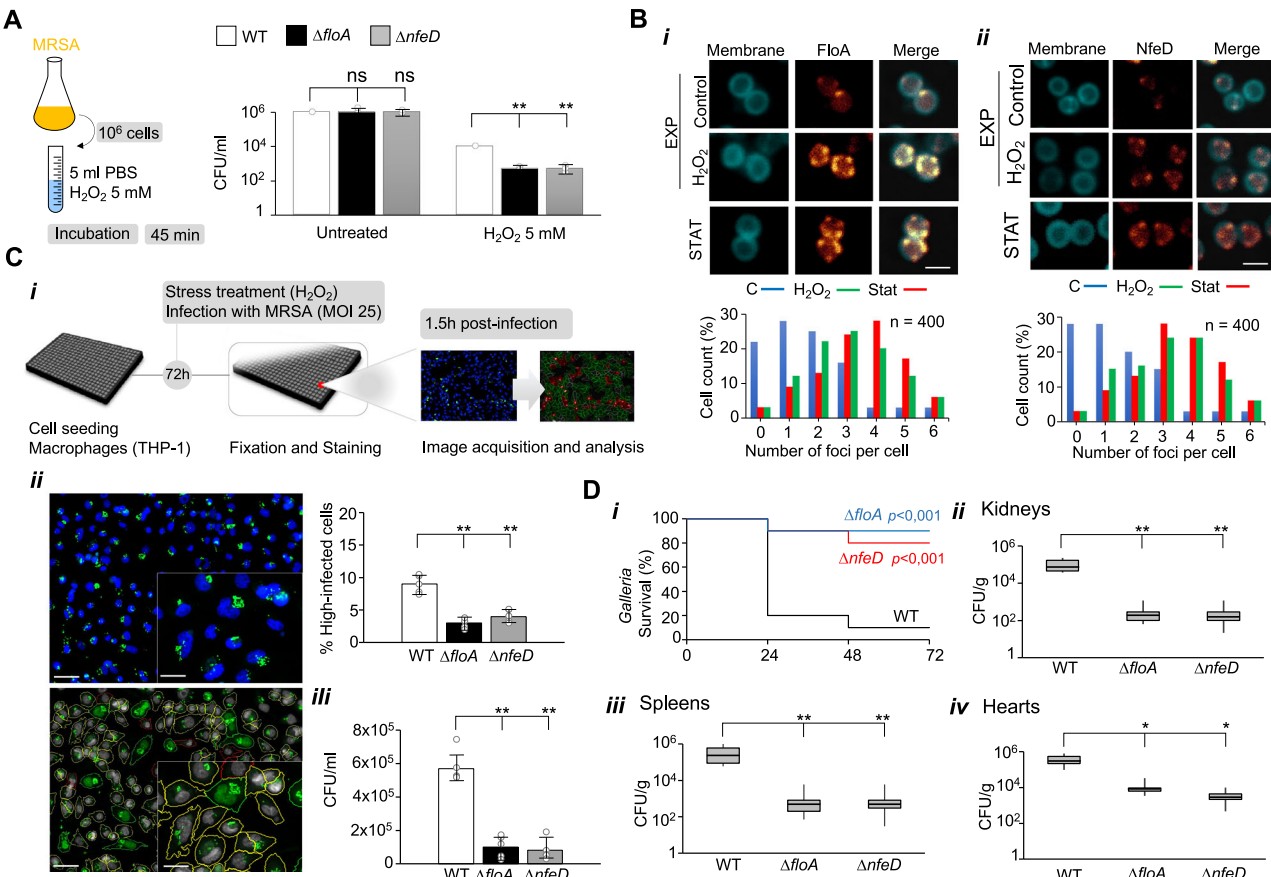

**Fig. 1 | FloA and NfeD are required for MRSA survival during infection.**
**A** Bacterial survival, measured as CFU/mL, of the different MRSA strains grown in TSB medium with or without 5 mM $H_2O_2$. Differences were measured by one-way ANOVA with the Tukey test for multiple comparisons; ** $p < 0.01$, ns = not significant. Data are shown as mean ± SD of three independent experiments ($n = 3$). **B** Upper panel: fluorescence microscopy micrographs of FloA-YFP (i) or NfeD-YFP (ii) labeled MRSA cells collected from untreated exponentially-growing cells (EXP control), $H_2O_2$-treated cultures at exponential (EXP $H_2O_2$) or stationary (STAT) phases of growth. Scale bar = 2 µm. Bottom panel: Quantitative determination of the number of fluorescence foci per cell in the labeled strains (sample of 400 cells). **C** MRSA intracellular replication in THP1 cells (MOI 25) at 1.5 h.p.i, evaluated by fluorescence microscopy-based infection assay and CFUs (i) Schematic representation of the infection workflow. (ii) Fluorescence microscopy image (upper panel) and corresponding image segmentation (lower panel) of THP1 cells infected with MRSA. In the original image, the nucleus and MRSA are shown in blue and green, respectively. In the segmentation panel, non-infected cells are outlined in red, infected cells with low and high bacterial intracellular levels are outlined in yellow and outlined/shaded in green, respectively (scale bar = 100 and 50 mm in the

uncropped and zoom image, respectively). Differences were measured by one-way ANOVA with the Tukey test for multiple comparisons ** $p < 0.01$. Data are shown as mean ± SD of three independent experiments ($n = 3$) (iii) Quantification of intracellular *S. aureus* by CFU. Differences were measured by one-way ANOVA with the Tukey test for multiple comparisons; ** $p < 0.01$. Data are shown as mean ± SD of three independent experiments ($n = 3$). **D** (i) in vivo infection assays using an invertebrate infection model. *Galleria mellonella* were infected with different MRSA strains ($10^6$ CFU). Surviving larvae were counted after 48 h of incubation ($n = 15$ larvae/group; 3 independent experiments). Differences in survival were analyzed by the log-rank test. (ii–iv) in vivo infection assays using a murine sepsis model. Mice were infected intraperitoneally ($n = 10$; $3 \times 10^7$ CFU) and infections were allowed to progress for two days before the infected organs were collected and CFUs counted. Quantification of bacterial loads in kidneys (ii), spleens (iii), and hearts (iv). Results were examined by one-way ANOVA with the Tukey test for multiple comparisons. Top and bottom of the box indicates the 75th and 25th percentile, respectively. Whiskers extend to 1.5 times the interquartile range from the box.
* $p < 0.05$, ** $p < 0.01$.

residues, partly homologous to a eukaryotic sterol-sensing domain (SSD) (hence referred to as SSD-like region SSDL)[58,59] (Fig. 2Ai, Supplementary Fig. 2B and Supplementary Movie 1). *S. aureus* membranes do not contain cholesterol but they have other isoprenoid lipids, such as STX, with similar physico-chemical properties to cholesterol. The SSDL region is linked to a cytosolic five-stranded closed beta-barrel that topologically resembles an OB-fold (OBL, OB-fold-like domain)[60,61] (Supplementary Fig. 2C), a common ligand-binding domain[62].

Although structurally distinct, both proteins were detected in an FMM-enriched fraction (a detergent-resistant membrane fraction DRM—a subfraction of the membrane enriched in FMM, as opposed to the detergent-sensitive membrane fraction DSM) (Fig. 2Aii). To determine whether this is due to specific recognition of FMM lipids, NfeD and FloA variants with altered lipid-interacting domains were generated, and their potential confinement in FMM examined.

The aromatic residues of SSD provide additional hydrophobicity to the transmembrane segments and higher affinity for the interaction with sterol[58,59]. Thus, a NfeD$_{SSDL}$ variant, altered in SSDL aromatic residues showed poor FMM confinement but rather a homogenous membrane distribution in MRSA cells (Fig. 2Bi). Consistently, a NfeD$_{SSDL}$-YFP construct showed a uniform fluorescent signal distributed in MRSA membranes (Fig. 2Bii), suggesting that the SSDL domain is required for NfeD localization at FMM. In FloA, the PHB domain is known to recognize the FMM lipid STX[29]. PHB variants with systematic deletions were purified and tested for STX recognition using a lipid flotation assay. In this assay, PHB variants and lipids were mixed beneath a sucrose gradient followed by ultracentrifugation. Lipids migrated to the low-sucrose-density fraction at the top of the tube accompanied by PHB variants only when interaction occurred. The variants lacking α-helices 1 and 2 in subdomain-1 could not bind STX (Fig. 2Ci–ii and

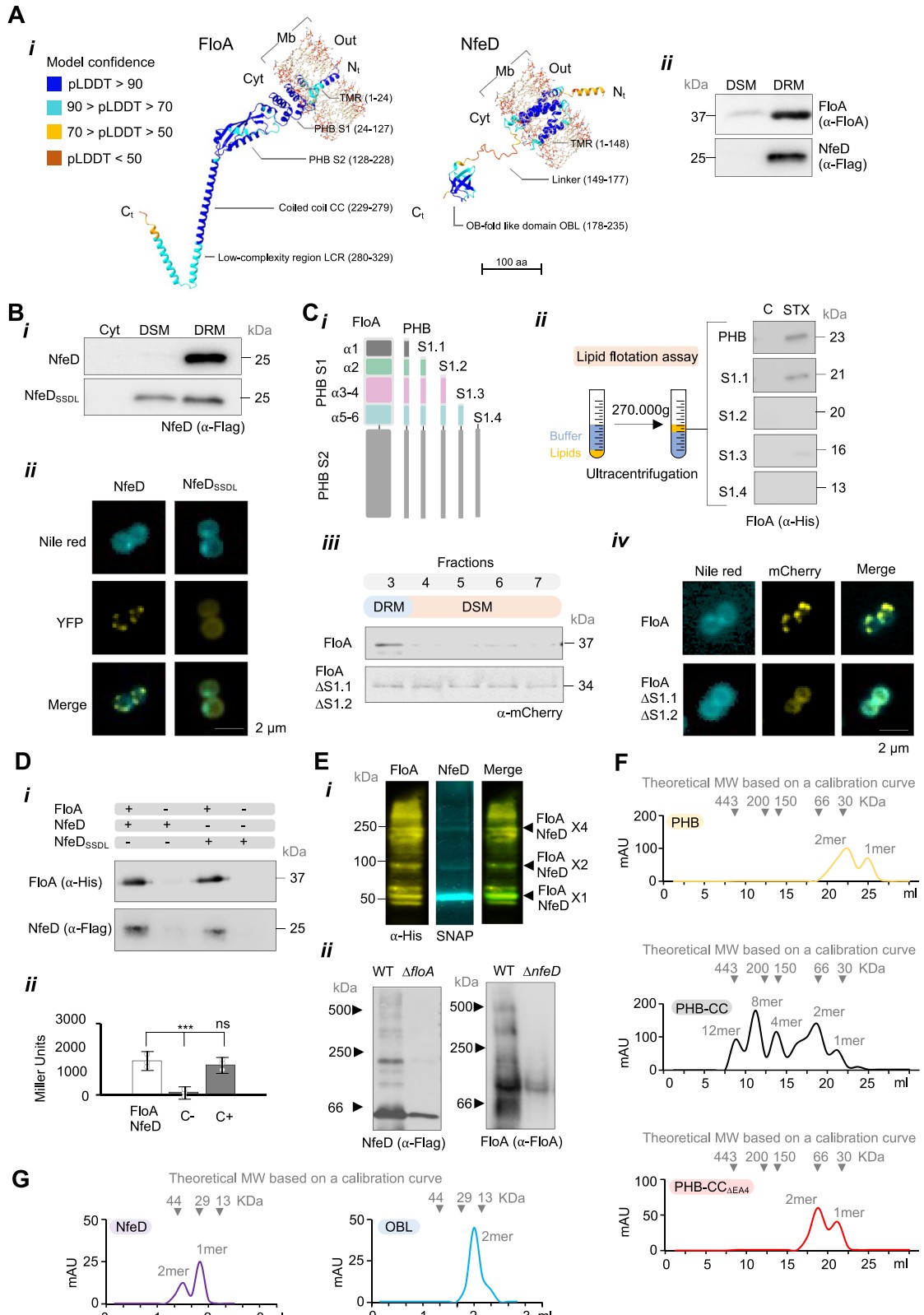

Supplementary Fig. 3A), showing a poor FMM confinement in MRSA cells using a membrane fractionation assay (Fig. 2Ciii). Consistently, in a fluorescence microscopy assay, a mCherry-labeled FloA lacking α-helices 1 and 2 distributed homogeneously in MRSA membranes (Fig. 2Civ), indicating that the subdomain-1 is required for FloA localization at FMM. In sum, FloA and NfeD follow specific lipid-recognition mechanisms for their accumulation at FMM.

Whether FloA and NfeD are functional partners when confined to FMM was examined by analyzing FloA-NfeD interaction occurring at these sites. Using pull-down experiments, purified FloA was immobilized in a Ni-NTA resin and incubated with MRSA extracts before FloA was eluted from the resin. Subsequent NfeD immunodetection in FloA co-eluted protein samples showed a FloA-NfeD interaction. A FloA-NfeD interaction was also detected when the pull-down assays

**Fig. 2 | FloA and NfeD recognize FMM lipids and concentrate in FMM to oligomerize. A** (i) AlphaFold2 structure prediction of flotillin FloA (left) and NfeD (right) colored by per-residue confidence score (pLDDT). TMR is transmembrane region. PHBS1 is PHB subdomain-1; S2 subdomain-2. CC is the coiled-coil region and LCR is the low-complexity region. In NfeD, the OBL is OB-like domain of NfeD. (ii) Immunodetection of FloA (37 kDa) or NfeD (26 kDa) in an FMM-enriched membrane fraction (DRM) and a phospholipid-enriched membrane fraction (DSM). **B** (i) Immunodetection of NfeD or NfeD$_{SSDL}$ variants in MRSA fractionated extracts; Cytoplasm Cyt or DSM, DRM membrane fractions. (ii) Fluorescence micrographs of NfeD-YFP or NfeD$_{SSDL}$-YFP in MRSA cells. Scale bar = 2 µm. **C** (i) Diagram showing the purified PHB variants lacking different PHB-S1 α-helices. (ii) Lipid flotation assay. After centrifugation, lipids migrate to the low-density sucrose fraction at the tube top. PHB was then immunodetected in the fractions at the tube top. C is a control assay using phosphatidylglycerol. STX is purified staphyloxanthin. (iii) Immunodetection of WT FloA and a FloA variant lacking α1-2 of PHB-S1 in MRSA extracts fractionated in a glycerol gradient. Cytoplasm Cyt or DSM, DRM membrane fractions. (*iv*) Fluorescence micrographs of FloA-mCherry or FloA$_{ΔS1α1-2}$-

mCherry in MRSA cells. Scale bar = 2 µm. **D** (i) FloA immunodetection in protein samples co-eluted with NfeD-Flag. The presence or absence of the prey (FloA) or bait (NfeD-Flag) is represented at the top of the panel with + or −, respectively. (ii) Quantification of FloA-NfeD interaction efficiency in a β-galactosidase assay using a bacterial two-hybrid analysis (B2H). Negative control (C−) is empty plasmids and positive control (C+) is Zip interaction. Data are shown as mean ± SD of three independent experiments (*n* = 3). **E** (i) FloA and NfeD dual labeling in SDS-PAGE of crosslinked MRSA membrane extracts. NfeD bands colocalize with FloA bands. FloA binds many client proteins and may generate additional bands of lower intensity. (ii) BN-PAGE and immunoblotting to detect FloA and NfeD oligomers in WT and mutants. High-MW oligomers are missing in the mutants. **F** Size-exclusion chromatography (SEC) profiles of soluble FloA variants. Top panel is PHB domain. The center panel is PHB domain + CC and LCR regions (PHB-CC) and the bottom panel is the PHB variant altered in the fourth EA repeat of the CC region (PHB-CC$_{ΔEA4}$). Arrows show protein standards for calibration. **G** SEC profiles of the native NfeD (upper panel) or purified OBL (lower panel).

were performed using NfeD$_{SSDL}$ extracts (Fig. 2Di). The FloA-NfeD interaction also occurred in a heterologous system using a bacterial two-hybrid assay (B2H) (Fig. 2Dii), pointing to a direct interaction between FloA and NfeD requiring no additional MRSA protein components. Using crosslinked samples, a co-detection of FloA and NfeD in several bands indicated the existence of distinct FloA-NfeD oligomerization states (Fig. 2Ei). The high-MW bands were compromised in the Δ*floA* or Δ*nfeD* mutants (Fig. 2Eii), highlighting a stabilizing role for NfeD in forming FloA oligomers.

Size-exclusion chromatography (SEC) was used to explore the distinct oligomerization states of purified FloA and NfeD (Fig. 2F, G and Supplementary Fig. 3B, C). The purified PHB domain of FloA resolved into monomers (30 kDa) and dimers (60 kDa), whereas a PHB-CC extended FloA variant generated additional higher-MW complexes (Fig. 2F). However, a PHB-CC variant harboring a deletion in the CC region (PHB-CC$_{ΔEA4}$, with A$_{286}$E$_{287}$A$_{288}$ deleted)[29] only resolved into monomeric and dimeric variants (Fig. 2F). These results indicated that FloA is arranged in dimers and multimeric complexes; FloA dimerization may be driven by a PHB-PHB interaction whereas FloA multimerization may involve the CC region. The SEC analysis also showed NfeD and the OBL domain to form dimers (Fig. 2G), suggesting that NfeD dimerization involves an OBL-OBL interaction. Taken together, these findings show that FloA and NfeD are recruited by FMM lipids and accumulate at FMM, where they interact and oligomerize to carry out their function.

## Flotillin oligomerizes in a clamp-shaped conformation

FloA and NfeD were subjected to structural analyses to understand the functional organization of flotillin at FMM. FloA and NfeD were co-produced in a heterologous system and purified by tandem affinity purification (TAP) (Supplementary Fig. 3Di–iii) followed by cryo-EM analysis (Fig. 3A, Supplementary Fig. 4, Supplementary Movie 2 and Supplementary Dataset 2). It is important to consider that, while the field of cryo-EM achieved remarkable technological advances, a persistent challenge lies in effectively resolving small proteins (below 100 kDa), such as FloA or NfeD. To date, only a few cases of small proteins have been successfully resolved by cryo-EM[63], primarily due to the low signal-to-noise ratio (SNR) in the images. To circumvent the low contrast and small size of our particles, we used manual picking for most of our samples and used no template at any step of particle picking (Supplementary Fig. 4). Using this approach, we obtained 3D volumes for FloA (118 K particles, 8.0 Å of resolution) and FloA-NfeD (41 K particles, 7.9 Å of resolution) (Fig. 3A). Comparable particles in shape and size were obtained in lower resolution maps using negative-staining EM (Supplementary Fig. 5A). In sum, despite the challenge to resolve small proteins using cryo-EM, we obtained 3D maps of the proteins with sufficient information to understand the role

of flotillin in its biological context. Despite this, further work would be necessary to obtain high-resolution structures to validate structural interpretations. In the 3D maps, The FloA monomer showed the PHB domain to be arranged into a plane facilitating interaction with FMM lipids. The CC region reached the cytoplasm and a short linker turned the LCR region upwards (Fig. 3Ai). In the FloA-NfeD particle, it was possible to confidently fit the AF2 predicted NfeD structure in the extra density present at the beginning of the CC-LCR flexible region (Fig. 3Aii, iii), which corresponded to the OBL domain of NfeD. The cryo-EM map revealed the interaction of the OBL and the CC region of FloA to cause the bending of the CC region, forming a nook space that accommodates the OBL, with the CC-LCR flexible region extended to the side as a flexible hook or tentacle. Based on this, a potential NfeD-FloA interaction interface was identified in the OBL domain (β3-β4) to interact with the CC region of FloA (Fig. 3Aiii and Supplementary Fig. 5B). To confirm the cryo-EM map, a NfeD variant (V1) altered in this OBL interaction interface was engineered, which did not generate an interaction signal with FloA in a B2H assay. However, alterations in other region of the OBL domain (V2) did not interfere with FloA interaction (Supplementary Fig. 5C).

The dimeric state of flotillin were separated by SEC[36,64] (Supplementary Fig. 3Div). Peak fractions were validated by negative-staining EM followed by examination by cryo-EM analysis. Images from peak fractions generated maps of FloA-NfeD (2XFloA-NfeD) (49 K particles) dimers at 8.8 Å resolution (Fig. 3B, Supplementary Fig. 6, Supplementary Movie 3 and Supplementary Dataset 2), which revealed the parallel assembly of two identical units, validated by fitting the AF2 Multimer interaction model into the cryo-EM maps. Particles with similar shape and size were obtained in lower resolution maps using negative-staining EM (Supplementary Fig. 7A). The FloA-NfeD monomers lied side by side and their PHB domains confronting one another. The CC-LCR tentacles protruded far from the base of the dimer (Fig. 3Bi). The dimerization occurred via PHB-PHB interactions (Fig. 3Bii), consistent with the SEC experiments and confirmed by chemical cross-linking followed by mass spectrometry[65]. We detected a MS fragmentation spectrum of a PHB-PHB linked fragment, in which the region near the end of the β2 strand in one PHB domain was close to the region adjacent to the end of the β3 strand in another domain (Fig. 3Biii and Supplementary Fig. 5D), consistently to the PHB-PHB interacting region described in a crystallographic study[57]. The FloA-NfeD dimer showed bent tentacles in a symmetrical hook-like arrangement, which allowed the dimer to adopt a clamp-shaped conformation.

Images that were collected from the peak fractions closed to the void volume of the SEC column showed heterogenous sample of ring-shaped multimeric particles, consistent with the topology of structurally-related proteins[66–70]. The multimeric particles were fragile,

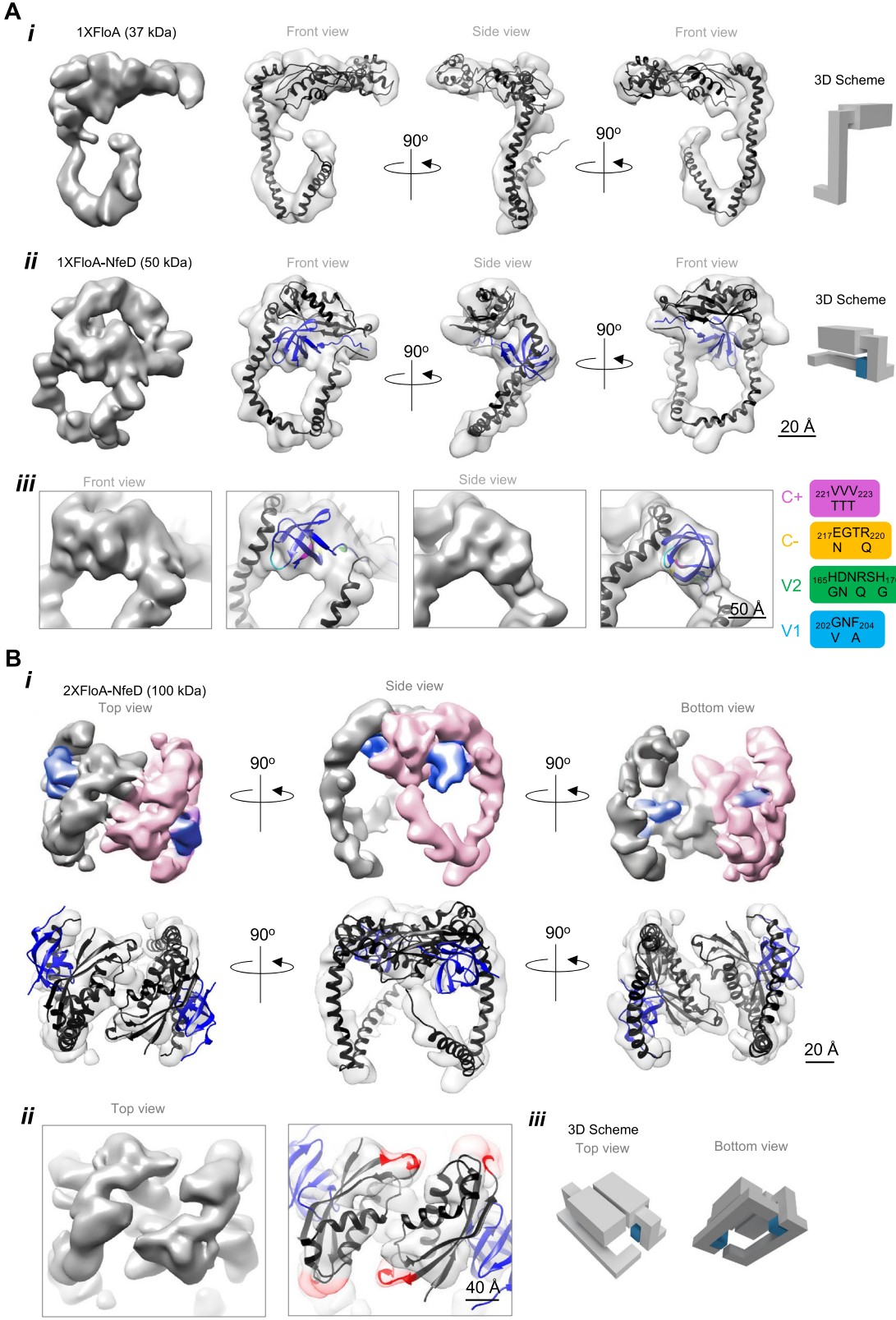

as treatments with lipid-disturbing agents, such as DDM (1%) or lipase (100 nM), caused their disaggregation (Supplementary Fig. 7B), pointing to the critical role of membrane lipids in their assembly. Nonetheless, a general structural organization of the multimeric ring-shaped particles was detected using negative-staining EM (Supplementary Fig. 7C). According to the SEC results (Fig. 2F, G), dimer-to-dimer interaction seemed to narrow down to the upper part

of the tentacle that is bound to the OBL domain, in agreement with the SEC results, suggesting that OBL-OBL interaction may connect two different dimers. Using a B2H assay, we detected a disruption of the OBL-OBL interaction in OBL variants with an altered β1-β2 interface (V2), whereas the interaction remained in the V1 variant (Supplementary Fig. 7D). In sum, a clamp-shaped dimer is the building block of the multimers. The multimeric assembly involves

**Fig. 3 | Cryo-EM structure of FloA and FloA-NfeD. A** (i) FloA cryo-EM map (left) and different views of the fitted AlphaFold2 atomic model (right). (ii) FloA-NfeD cryo-EM map (left), and different views of the fitted AlphaFold2 atomic model (right). Schematic representation of FloA and FloA-NfeD organizational structure are shown on the right. (iii) Zoomed-in views of the EM density for the area of FloA that binds the OB-fold of NfeD. Cryo-EM map is shown in the left panel and the fitted AlphaFold2 atomic model is shown in the right panel. The residues altered in the OB-fold by site-directed mutagenesis are colored and described on the right. **B** (i) Cryo-EM map of the FloA-NfeD dimer with flexible fitting of the AF2 Multimer predicted model for the FloA-NfeD dimer. Different views of the FloA-NfeD dimer

cryo-EM map. One FloA monomer is labeled in pink and the other FloA monomer in gray. The OBL of NfeD is shown in blue. In the FloA-NfeD dimer, the OBL caused the bending of the flexible CC-LCR tentacles to position them laterally in a clamp-shaped conformation. (ii) Zoomed-in views of the EM density for the area of PHB-PHB dimerization. The cryo-EM map is shown in the left panel and the fitted AlphaFold2 atomic model is shown in the right panel. The proximal regions in the dimer that were detected by DSS cross-linking are marked in red. (iii) Schematic representation of 2xFloA-NfeD organizational structure. FloA is represented in gray whereas the OB fold of NfeD is represented in blue.

---

both the interaction with membrane lipids and a dimer-to-dimer interaction mediated by NfeD.

### FloA-NfeD stabilizes unfolded proteins at FMM

The two hooked flexible tentacles of the FloA-NfeD dimeric unit opened a cavity accessible to any structural element. Protein structures with CC-OB tentacles are known to prevent the aggregation of denatured proteins[62,71–73] (Supplementary Fig. 7E). These flexible tentacles mediate non-specific recognition of solvent-exposed hydrophobic surfaces in unfolded proteins[74,75], whereas the OB-fold provides stability to the complex. The tentacles bind to and stabilize unfolded proteins and upon stabilization, refolding of the client protein to its native state becomes energetically favorable and likely to occur without CC-OB tentacles playing any direct role in protein refolding[76]. We thus hypothesized that flotillin assemblies are purposely dedicated to stabilizing FMM client proteins to favor their correct folding.

The hypothesis above implies that FMM client proteins are likely unfolded proteins. As FMM are ordered membrane regions with reduced fluidity, like other raft-like membrane microdomains, water molecules are excluded and their hydration level is low[77]. Being FMM highly hydrophobic membrane regions, they may provide a stabilizing environment to unfolded proteins that become insoluble in the aqueous cytoplasm, as they expose their hydrophobic side-chains to the solvent[78]. FMM could contribute to isolating unfolded proteins from the rest of the cell to prevent protein insolubility and precipitation thus, we investigated the insolubility of FMM client proteins compared to the rest of the membrane proteins. Protein insolubility does not strictly equate to protein unfolding, but changes in proteome solubility is usually associated with an altered proteome stability[79]. To this end, proteins from an FMM-enriched membrane fraction (DRM) and a phospholipidic membrane fraction (DSM) of stressed cells ($H_2O_2$ 5 mM) were purified and solubilized in aqueous buffer, followed by centrifugation to precipitate insoluble proteins (20.000 g, 45 min, 4 °C) (Fig. 4A). SDS-PAGE showed the different protein content of DRM and DSM fractions (Fig. 4A). The DSM protein content was similar in the total and the soluble protein fractions and consistently, the insoluble protein fraction rendered a low signal. In contrast, the DRM showed a drop in the protein content of the soluble protein fraction compared to the total protein fraction. Accordingly, the insoluble fraction showed an important protein signal, suggesting that a large fraction of DRM proteins were insoluble and precipitated. To test the contribution of FMM lipids to the recruitment of insoluble proteins to the FMM, we used gradient fractionation to separate the FMM-enriched fraction (fractions 2 and 3) from the rest of the membrane. FloA was detected in these fractions (Fig. 4B), whereas a mutant defective in STX biosynthesis (Δcrt) showed FloA distributed homogenously across the cellular membrane. By quantifying soluble and insoluble proteins in these fractions, we detected that stressed WT cells consistently accumulated insoluble proteins in the FMM-enriched fraction, whereas the Δcrt mutant showed a uniform distribution of the insoluble proteins across the entire cellular membrane (Fig. 4B). These results indicated that insoluble proteins accumulate at FMM. Additionally, the ΔfloA and ΔnfeD mutants showed more insoluble proteins accumulating at FMM than WT (Fig. 4C), suggesting that

insoluble proteins accumulated in FMM in stressed cells and flotillin contributes to their stabilization.

To identify the insoluble proteins that are harbored in the FMM, MS-based protein identification was performed at different points over growth in a chemically defined poor medium that caused bacteria to starve and grow stressed[80] (Supplementary Dataset 3). We monitored STX production to assess cellular stress, as the production of this pigment is triggered by the stress-induced σB[81,82]. The DRM/DSM relative protein concentration was represented in a heat map (Fig. 4Di). A core of ~200 diverse proteins was consistently enriched at FMM at all points over the growth curve, whereas other proteins accumulated at FMM during exponential or stationary phase. We found a highly diverse group of proteins in FMM (Fig. 4Dii) (e.g., PBP2a is a FloA-interacting protein in MRSA in its insoluble state, as PBP2a faces the extracellular side of the membrane in its correct folded state)[29]. This led us to hypothesize that insoluble proteins at FMM are diverse and FMM may stabilize these proteins to undergo their cellular localization pathway and correct folding. To explore whether proteins at FMM are insoluble proteins in the absence of flotillin, WT and ΔfloA mutant were grown in chemically defined poor medium[80], in which bacteria grew stressed; the protein synthesis of cultures was blocked (chloramphenicol 10 µg/ml), followed by a heat shock treatment to cause protein unfolding (55 °C, 30 min) and further incubation (37 °C for 30 min) to allow unfolded proteins to refold. Samples were collected at each step, their membrane fraction resuspended, and the insoluble proteins were precipitated and quantified in relation to soluble proteins (Fig. 4Ei). We detected a significant increase in protein insolubility after the heat shock treatment. However, protein insolubility became more pronounced after incubation at 37 °C only in ΔfloA, whereas WT recovered protein solubility upon incubation (Fig. 4Ei), pointing to the crucial role of flotillin in favoring the solubility of client proteins. We used MS-based proteomics to identify insoluble proteins in the ΔfloA mutant (Fig. 4Eii–iv and Supplementary Dataset 4). The most abundant insoluble proteins detected in ΔfloA were previously detected as FMM client proteins (>90% of the total). In contrast, underrepresented insoluble proteins were rarely detected in FMM (>90% of the total), indicating that most FMM-accumulating proteins become unstable and precipitated in the absence of flotillin.

To investigate whether insoluble proteins accumulated at FMM without the assistance of additional components other than the hydrophobic affinity, we designed an in vitro assay in which MRSA protein samples were forced to unfold by heat shock treatment (65 °C, 15 min), followed by their incorporation into liposomes containing STX and phosphatidylglycerol (PG) (Fig. 4Fi) (ratio 1:5). The liposome lipids segregated laterally into regions with different hydrophobicity, as monitored by fluorescence microscopy using C-Laurdan staining[83] (Fig. 4Fii). The STX and PG membrane fractions of the liposomes were isolated by membrane fractionation and their protein fractions purified, followed by aqueous solubilization and precipitation of insoluble proteins by centrifugation. Insoluble proteins behaved similarly regardless of whether they had a membrane or cytoplasmic origin, and both accumulated in the STX fraction of the liposomes compared to the PG fraction (Fig. 4Fiii). In contrast, the untreated samples showed a reduced concentration of insoluble proteins and a more homogenous

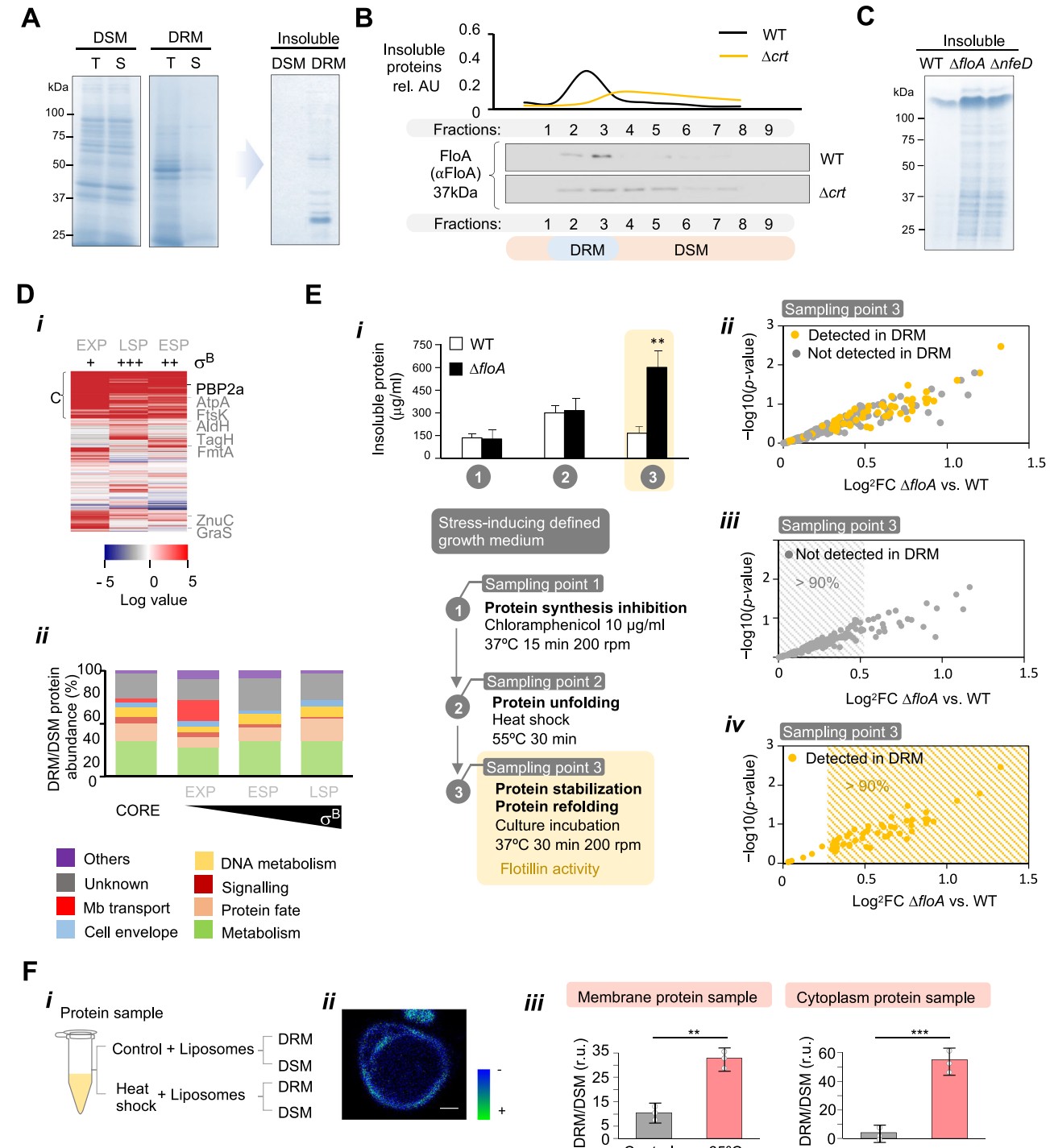

distribution in the liposomes. In sum, in vitro and in vivo assays showed that the particular lipid composition of FMM leads to the accumulation of insoluble proteins at FMM. Importantly, the in vitro assays showed that this occurred without the addition of ATP or any other energy source to the assay, suggesting that the recruitment of insoluble proteins at FMM in the cell is mainly driven by hydrophobic interactions without any direct ATP cost.

Upon recruitment of insoluble client proteins at FMM, the flotillin clamp may hold the client proteins to stabilize them and upon stabilization, it is likely that these proteins undergo their correct folding without further flotillin assistance. We investigated this hypothesis using the FMM client protein PBP2a as example because purified PBP2a is prone to unfold under minor stress perturbances[84] and it became

insoluble in ΔfloA and ΔnfeD mutants (Supplementary Fig. 8Ai). Thus, purified PBP2a was incubated at increasing temperatures followed by separation into soluble and insoluble PBP2a by centrifugation (20.000 × g, 45 min, 4 °C) (Fig. 5Ai). We detected a clear onset of PBP2a insolubility at 42 °C, with the PBP2a unfolding and precipitating. However, adding purified FloA (1:1 ratio) to the assay caused PBP2a to remain soluble, even at temperatures higher than 42 °C. An increase in PBP2a solubility was detected when purified FloA-NfeD was added to the reaction. In contrast, adding NfeD alone did not stabilize PBP2a and precipitated at 42 °C or higher temperatures. The transmembrane region did not affect PBP2a solubility. A purified PBP2a$_{\Delta TMR}$ variant that lacked the transmembrane region also became insoluble and precipitated at 42 °C and higher temperatures, but it remained soluble

**Fig. 4 | FMM recruits client proteins in their unfolded state. A** Left, SDS-PAGE of total (T) and soluble (S) protein fractions of DRM and DSM of MRSA cells. Right, SDS-PAGE of the insoluble protein fraction of DRM and DSM of MRSA cells. The insoluble protein fraction has been concentrated compared to the total and soluble fractions. **B** Relative amount of insoluble proteins in DRM fraction compared to the DSM fraction. The insoluble proteins concentrate in fractions 2 and 3 of the gradient, where most FloA signal localizes. The Δ*crt* mutant shows a homogenous distribution of FloA and insoluble proteins in DRM and DSM. **C** SDS-PAGE of insoluble proteins in standardized cell extracts of different MRSA mutants. **D** MS-based quantification of the DRM and DSM proteome in the exponential (EXP), late stationary phase (LSP) and early stationary phase (ESP) of growth. (i) Heat map showing the protein abundance ratio in DRM and DSM fractions (determined via unsupervised hierarchical clustering). Red denotes a DRM increase relative to DSM, and blue a reduction. Gray boxes indicate missing values. PBP2a and other client proteins are highlighted. C indicates the core proteins. (ii) Functional classification of DRM proteins, according to the TIGRFAMM database. **E** (i) Upper panel: Quantification of insoluble proteins vs. soluble proteins upon inhibition of protein synthesis (sampling point 1), heat shock treatment (sampling point 2) or incubation

at 37 °C (sampling point 3). Results were examined by Student's *t* test; **$p < 0.01$. Data are shown as mean ± SD of three independent experiments ($n = 3$). Bottom panel: Diagram of the in vivo assay for the accumulation of unfolded proteins at the FMM. (ii) MS-based identification and analysis of the insoluble proteins at sampling point 3 in Δ*floA* vs. WT. The proteome database was cross-checked with the proteome of the FMM-enriched fraction (DRM) (**D**). The most represented insoluble proteins in Δ*floA* were detected as FMM proteins (>90% of the total) (iii), whereas low-represented insoluble proteins were rarely detected in FMM (>90% of the total) (iv). **F** (i) Diagram of the in vitro assay for the accumulation of insoluble proteins at the FMM. Control native or previously insoluble proteins were embedded in STX-PG containing liposomes and the protein distribution pattern was analyzed. (ii) Confocal fluorescence microscopy image of C-Laurdan staining of STX-PG containing liposomes. C-Laurdan reports membrane regions with greater hydrophobicity by switching fluorescence from blue (−) to green (+). Scale bar is 10 μM. (iii) Quantification of insoluble proteins associated with the STX and PG fractions of the liposomes. Differences were measured by Student *t* test; **$p < 0.01$, ***$p < 0.001$. Data are shown as mean ± SD of three independent experiments ($n = 3$).

in the presence of FloA (Supplementary Fig. 8Aii). Collectively, this in vitro assay showed the capacity of flotillin to stabilize unfolded PBP2a without the addition of ATP or any other energy source.

The capacity of PBP2a to fold correctly upon FloA stabilization was determined by monitoring the PBP2a transpeptidase activity. During peptidoglycan synthesis, peptidoglycan glycosyltransferases (PGTs) catalyze the polymerization of the peptidoglycan precursor Gly$_5$-Lipid II into glycan strands whereas the transpeptidase activity of PBP2a crosslinks the resulting glycan strands (Supplementary Fig. 8B, C). Thus, purified PBP2a was incubated with FloA-NfeD under stabilizing (37 °C) or unfolding (42 °C) temperatures. After incubation, previously-purified Gly$_5$-Lipid II and a PGT (PBP$^{S398G}$, a PBP with an active PGT domain and an inactive transpeptidase domain)[85] were added to the reaction to allow peptidoglycan polymerization. The synthesized peptidoglycan was digested to the constituent muropeptides, followed by LC/MS analysis to detect the crosslinked muropeptides resulting from PBP2a transpeptidase activity (Fig. 5B and Supplementary Fig. 8D–F). At 37 °C, we detected the presence of crosslinked muropeptides (1209.0617 ([M + 2]/2)) in all samples. However, the detection of the crosslinked muropeptides at 42 °C only occurred when purified FloA was added to the reaction (FloA or FloA-NfeD). These results indicate that flotillin stabilizes insoluble client proteins and the stabilization lead the client proteins to refold correctly without further FloA assistance.

**Flotillin displays a clamp-shaped mechanism to stabilize client proteins**
To investigate the role of flotillin tentacles in the stabilization of unfolded proteins, FloA-NfeD were incubated with unfolded PBP2a and resolved by SEC (Supplementary Fig. 3Dv) prior examination by cryo-EM (Fig. 5C, Supplementary Fig. 9, Supplementary Movie 5 and Supplementary Dataset 2). We obtained structures corresponding to dimeric particles of FloA-NfeD (2XFloA-NfeD + PBP2a) (53 K particles) at 8.1 Å resolution. The volumes obtained showed the tentacles protruding from the base of the dimer in the typical clamp-shaped conformation, with an additional density between the tentacles attributable to monomeric PBP2a. The FloA-NfeD monomers lied side by side and their PHB domains confronting one another, comparable to the organization of the apo version of the clamp (Fig. 5C). In contrast, the hooked tentacles showed a slightly different disposition likely to embrace the PBP2a volume and to provide client protein stabilization (Fig. 5C). The density of PBP2a showed structurally unassigned regions, consistent with the unfolded state of PBP2a. A comparable outcome was obtained in lower resolution maps using negative-staining EM (Supplementary Fig. 7A). Using negative-staining EM, we were able to visualize the FloA dimer (2XFloA and 2XFloA +

PBP2a), which resulted less stable without NfeD and precluded cryo-EM analysis. Volumes revealed that PBP2a was captured with limited contact with the two tentacles (Supplementary Fig. 10A).

In vivo, the contribution of the tentacles to the stability of client proteins was tested in MRSA using a Δ*floA* strain complemented with a FloA variant that lacked the CC-LCR region (Δ*floA*, FloA$_{ΔCC}$). Compared to a Δ*floA* strain complemented with a WT copy of FloA (Δ*floA*, FloA), H$_2$O$_2$-stressed cultures of FloA$_{ΔCC}$ showed a more significant accumulation of unfolded proteins (Fig. 6Ai and Supplementary Fig. 10B), comparable to that seen for the Δ*floA* mutant. To test the PBP2a folding status in these strains, we grew them in the presence of β-lactam (oxacillin 6 μg/ml). The growth of the Δ*floA*, FloA$_{ΔCC}$ was severely reduced compared to Δ*floA*, FloA (Fig. 6Aii), suggesting that PBP2a integrity is compromised in MRSA in the absence of a FloA with functional tentacles.

**FloA-NfeD disassembly causes attenuation of multi-drug resistant MRSA infections**
FMM recruit and stabilize or refold proteins damaged by cellular stress, and consequently the inhibition of FMM activity compromises bacterial viability during infection (Fig. 1). As such, FMM represent outstanding targets against MRSA infections that cannot be treated with conventional antibiotics. FMM lipid biosynthesis in *S. aureus* can be impaired using inhibitors of the mevalonate pathway (MEV)[86], some of them marketed as cholesterol-lowering drugs in humans (i.e., statins) (Supplementary Fig. 10C). These have been shown to inhibit PBP2a activity in the strain USA300 (oxacillin MIC = 16 μg/ml) via an unknown mechanism[29,87]. We hypothesized that the FMM disassembling action of MEV inhibitors compromises proteome stability in infecting bacteria. The consequences of FMM inhibition in bacteria during infection were therefore explored in a collection of multi-resistant MRSA clinical isolates ($n = 13$) known for their prevalence in hospitals and their complex antibiotic resistance pattern (oxacillin MIC > 128 μg/ml) (Fig. 6B, Supplementary Fig. 10D and Supplementary Datasets 5 and 6). To test this, cells from H$_2$O$_2$-treated cultures with or without the MEV inhibitor zaragozic acid (ZA, 10 μM) were collected, their membrane fraction purified, and the relative amount of insoluble proteins was quantified (Fig. 6Ci). ZA-treated cultures of all isolates showed a higher accumulation of insoluble proteins in their membranes in response to H$_2$O$_2$ than did untreated cultures, which occurred concomitantly to a significant decrease in bacterial survival (Fig. 6Cii). As we showed that FMM are necessary to prevent PBP2a from unfolding during oxidative stress, MEV inhibitors may additionally cause PBP2a to unfold in these MRSA isolates that show higher levels of β-lactam resistance than USA300. The addition of oxacillin caused no significant growth differences in the MRSA isolates, but the

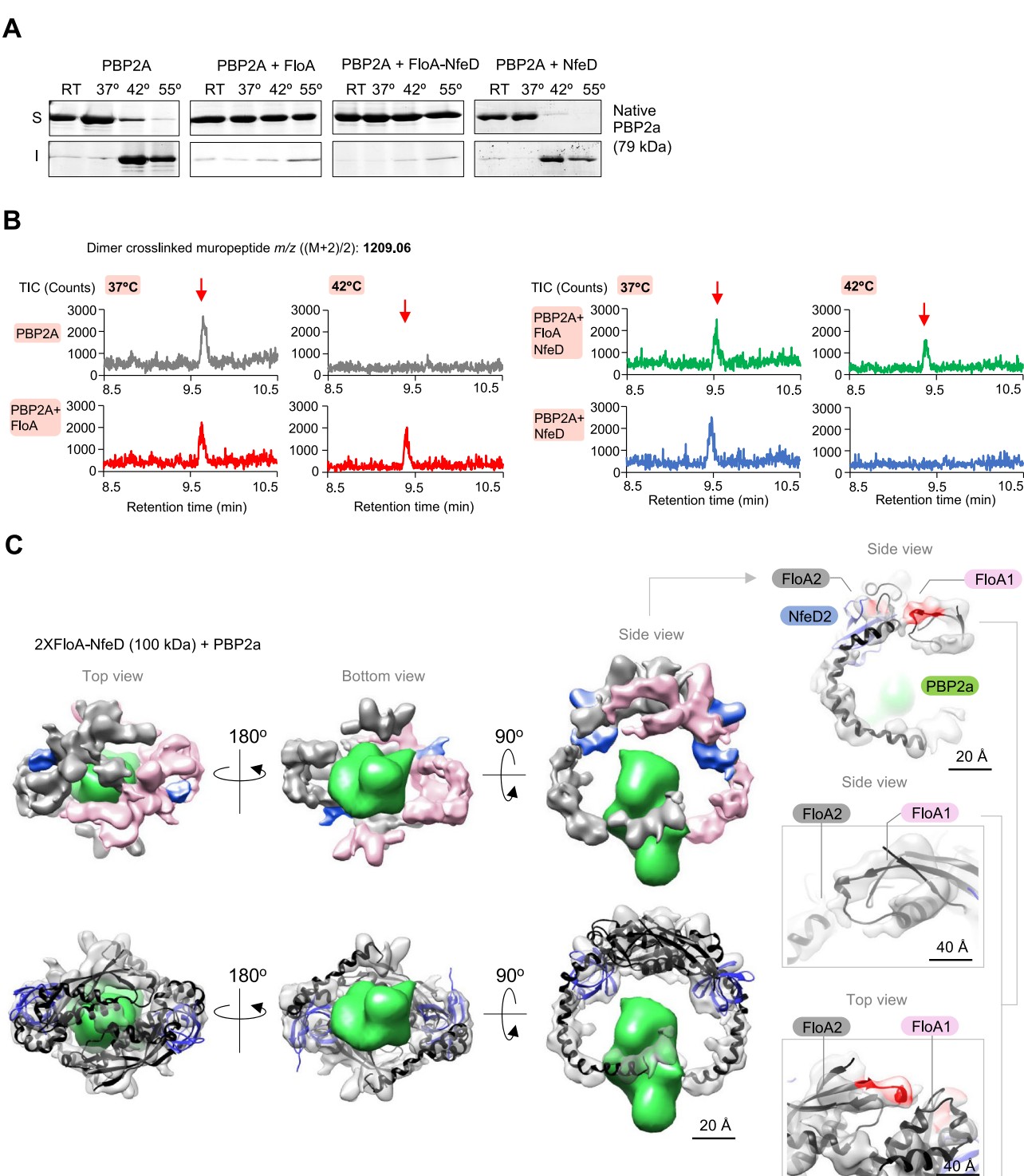

**Fig. 5 | Flotillin stabilizes unfolded PBP2a to prevent its aggregation. A** Thermal aggregation assay of PBP2a conducted at various temperatures. PBP2a shows a strong onset of insolubility at 42 °C. The addition of FloA or FloA-NfeD to the assay reduced PBP2a insolubility. As a control, adding NfeD alone did not reduce PBP2a insolubility. **B** PBP2a in vitro transpeptidase assay at different temperatures: LC-MS extracted ion chromatogram using purified PBP2a and Gly5-Lipid II with FloA and/or NfeD at 37 °C or 42 °C. The production of the crosslinked dimeric muropeptide at 42 °C only occurred in the presence of FloA or FloA-NfeD. **C** Cryo-EM map of the FloA-NfeD dimer bound to unfolded PBP2a. Flexible fitting of the AF2 Multimer prediction is shown in the bottom panels. The top panels show different views of the cryo-EM map, in which one FloA monomer is labeled in pink and the other FloA monomer in gray. The OBL of NfeD is labeled in blue. The PBP2a volume showed structurally undefined regions, as was in its unfolded state. As the tentacles of FloA-NfeD were extended laterally, they embraced the PBP2a volume, covering a large fraction of PBP2a. Right panels show different zoomed-in views (side and top views) of the EM density and the and the fitted AlphaFold2 atomic model for the FloA-NfeD dimer loaded with unfolded PBP2a. The proximal regions detected by DSS cross-linking are marked in red.

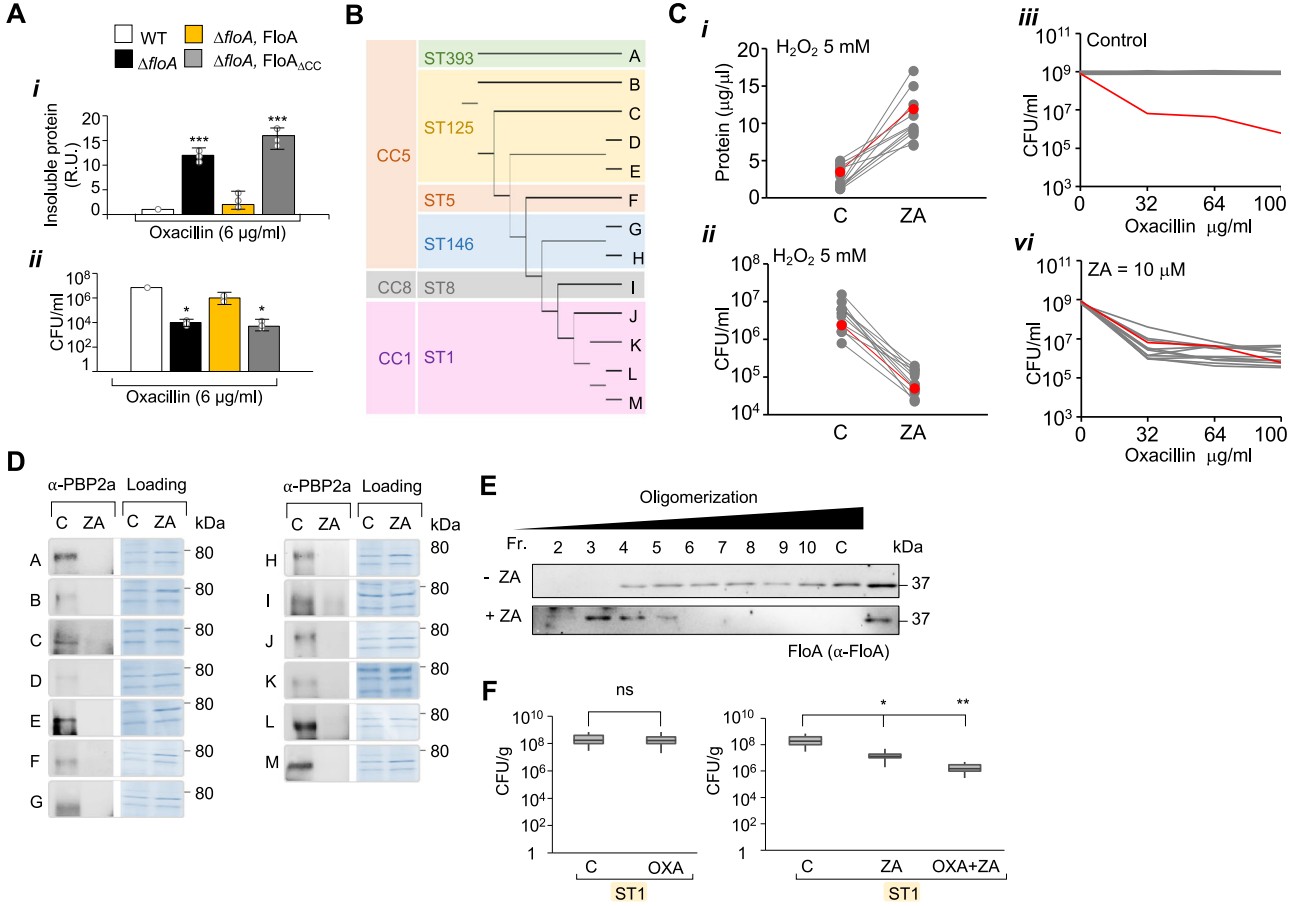

**Fig. 6 | FMM disruption affects flotillin oligomerization and leads to the accumulation of misfolded proteins.** **A** (i) Quantification of insoluble proteins associated with the membrane fraction from WT (white), the $\Delta floA$ mutant (black), the $\Delta floA$ complemented strain that produces $FloA_{WT}$ (yellow), and a $\Delta floA$ producing the $FloA_{\Delta CC\text{-}LCR}$ variant. Results were examined by one-way ANOVA with Tukey test for multiple comparisons; ***$p < 0.001$. Data are shown as mean ± SD of three independent experiments ($n = 3$). (ii) Quantification of β-lactam resistance as a proxy for PBP2a activity, measured as MRSA growth (CFU/ml) in the presence of oxacillin (6 µg/ml) in WT, $\Delta floA$, $\Delta floA$ producing $FloA_{WT}$, and $\Delta floA$ producing $FloA_{\Delta CC\text{-}LCR}$. Results were examined by one-way ANOVA with Tukey test for multiple comparisons; *$p < 0.05$. Data are shown as mean ± SD of three independent experiments ($n = 3$). **B** Phylogeny of multi-drug resistant isolates based on multi-locus sequence typing (ST). The classification shows three clonal complexes (CC): CC1, CC8 and CC5. **C** (i) Quantification of non-soluble proteins in the DRM and DSM fractions in cultures treated with $H_2O_2$, with or without zaragozic acid (ZA = 10 µM). In red is represented the benchmark clinical isolate USA300. (ii) Bacterial survival (CFU/mL) of $H_2O_2$-treated cultures in the presence or absence of ZA (10 µM). The lines connect non-treated and treated dots of the same strain. Data show the mean of three biological replicates. In red is represented the benchmark clinical isolate USA300. (iii–vi) Effect of MRSA resistance to oxacillin in the absence (iii) and presence (vi) of ZA (10 µM). In red is presented a benchmark control strain sensitive to oxacillin (Newman strain). **D** Immunodetection of soluble, functionally active PBP2a in the membrane extracts of ZA-treated and untreated cultures of clinical isolates (strains A to M) that are represented in **C**. **E** Glycerol gradient analysis of FloA oligomerization pattern in untreated (upper panel) or ZA-treated (bottom panel) cultures of a clinical isolate belonging to ST1 (Isolate L). **F** Bacterial load in lungs of oxacillin-treated infected mice in a pulmonary infection model. The left panel shows ST1-infected mice with or without oxacillin treatment. Oxacillin does not affect mice infection by ST1. The right panel shows ST1-infected mice treated with ZA alone or ZA and oxacillin. C is untreated infected mice. ZA treatment caused a reduction of bacterial load, which increased significantly in the presence of oxacillin. Mice were infected with $3 \times 10^8$ CFU ($n = 10$). One day after bacterial challenge, the lungs were collected and CFUs counted. Differences were analyzed by one-way ANOVA. Top and bottom of the box indicates the 75th and 25th percentile, respectively. Whiskers extend to 1.5 times the interquartile range from the box. *$p < 0.05$, **$p < 0.01$.

addition of oxacillin to ZA-treated (ZA, 10 µM) cultures caused severe growth inhibition (Fig. 6Ciii, vi and Supplementary Fig. 10D). MRSA re-sensitization to β-lactams was correlated to the amount of active PBP2a available as determined by immunoblotting. An important reduction of soluble PBP2a was detected in all ZA-treated cultures (Fig. 6D).

We selected the MRSA isolate ST1-t127 (Isolate L in Fig. 6B. multi-resistance strain with 12 resistance genes), a spread clone in Europe, Australia, and the USA[88] to test the effect of ZA in flotillin disassembly. The membrane fraction of $H_2O_2$-treated cultures with or without ZA was collected and FloA oligomerization was determined by glycerol gradient fractionation (Fig. 6E). ZA-untreated samples showed the FloA signal to be concentrated in high-density glycerol fractions indicative of protein oligomerization, whereas ZA-treated

samples showed the signal in low-density glycerol fractions, highlighting a deficient FloA oligomerization pattern. The importance of flotillin disassembly in ST1 was assessed in vivo using a pneumonia infection mouse model (Fig. 6F). Infected mice were treated with oxacillin (200 mg/kg) or with a combination of oxacillin (200 mg/kg) and ZA (50 mg/kg). Infections were allowed to progress for 24 h before lungs were collected and bacterial load determined. The oxacillin treatment alone caused no attenuation of the infection contrasting to ZA-treated mice, which showed a significant reduction of bacterial load (-12-fold reduction). The antibiotic adjuvant properties of ZA were detected in mice treated with oxacillin and ZA, in which a more severe reduction of the bacterial load (-100-fold reduction) was observed. Overall, FMM are crucial for bacterial survival against infection stress and targeting these sites can fight

multi-drug resistant infections that cannot be treated with conventional antibiotics.

## Discussion

The spatial compartmentalization of constituent proteins and lipids into microdomains is a general feature of all biological membranes. However, the contribution of these microdomains to specific cellular functions is still unclear[89,90]. Here, we show FMM in the human pathogen MRSA to confine proteins that become insoluble upon oxidative stress. Insoluble proteins, including unfolded proteins, are stabilized in FMM and prevented from circulating, thus avoiding unwanted interactions. At FMM, flotillin and NfeD form a clamp-shaped complex that recognizes unfolded proteins and stabilizing them, enabling them to acquire their correct conformation without flotillin playing any direct role in protein refolding (i.e. it is unlikely that flotillin shows foldase activity). Our results showed that the accumulation of unfolded proteins at FMM and their flotillin-based stabilization are both ATP-independent processes mainly driven by hydrophobic affinity, as both processes were reproduced in vitro with no addition of ATP or another energy source.

FMM may thus organize a cellular compartment for ATP-independent proteome repair that is essential for bacterial viability in conditions of ATP depletion, such as during an infection. When the immune system detects the presence of a pathogen, the influx of neutrophils to the infection site is accompanied by the production of ROS[91,92]. ROS inhibit bacterial growth by affecting protein folding and inhibiting the TCA cycle and the electron transport chain[93,94]. Bacterial respiration cannot, therefore, proceed as usual. The accumulation of neutrophils also leads to massive host cell oxygen consumption[95], further interfering with bacterial respiration and compromising the bacterial ATP levels. Therefore, the network of ATP-utilizing chaperones and proteases is severely compromised in bacteria depleted of ATP[78,96,97]. Under these conditions, FMM may play a vital role in bacterial survival to maintain the integrity of the bacterial proteome. This could explain the crucial role in infection of the *floA-nfeD* operon in MRSA and why this operon is conserved across almost all bacterial genomes[48–50]. Although FMM are structurally akin to the raft-like membrane microdomains of eukaryotes, whether eukaryotic membrane microdomains show any functional resemblance to that of bacterial FMM is unclear. However, it is known that alterations in eukaryotic microdomains or flotillin are typically related to the occurrence of neurodegenerative diseases, as well as other diseases related to protein misfolding and aggregation[98,99]. The present work could be instrumental in understanding the fundamental basis of disease caused by lipid raft malfunction in eukaryotic cells.

Flotillin is a conserved protein from bacteria to mammals and is known to be critical for the correct functioning of many cellular processes[100], but its precise role in cellular physiology is unknown. This work shows that flotillin holds insoluble proteins to stabilize them and prevent their aggregation These stabilized proteins are more prone to acquire their correct conformation and to fold correctly without needing further assistance. Flotillin incorporate FMM lipids and multimerize, similar to that seen for related proteins[66–68]. This multimeric assembly may provide additional advantages to flotillin activity, such as a fine regulation or higher stability during harsh stress conditions[101]. The NfeD protein plays a unique role in bacterial flotillin oligomerization. NfeD binds to flotillin tentacles and favors the stability and capture of unfolded proteins. This highlights the conserved role of NfeD in stabilizing bacterial flotillins and may explain why NfeD homologs are widely distributed in prokaryotes. However, no NfeD homolog has been described up to date in eukaryotes[48–50], suggesting that eukaryotic flotillin have evolved to dispense with the NfeD stabilizing function; an interesting differentiating aspect from a mechanistic and a functional perspective that can be further exploited to develop antimicrobials against bacterial infections.

Our results show that the pool of client proteins accumulated at FMM are mostly insoluble proteins, which includes unfolded proteins. Hence, FMM client proteins should vary between species and growth conditions and will combine nascent polypeptides that need the stabilizing role of FloA (e.g. PBP2a) with already-folded proteins that became unfolded as a consequence of stress[78,102–104] and folded proteins that become insoluble in these particular growth conditions and migrate to FMM. This explains the functional diversity of FMM-associated proteins[29,105], and why certain proteins are typically found in FMM to exclude others. In this regard, FMM disassembly using cholesterol inhibitors leads to the accumulation of unfolded proteins in MRSA membranes upon exposure to oxidative stress, affecting bacterial survival and attenuating infections of clinical isolates that cannot be treated with conventional antibiotics. These results are consistent with the increasing number of publications showing the beneficial role of cholesterol inhibitors in preventing staphylococcal infections[86,106–109]. These results also pointed to the additional effect of these inhibitors in β-lactam re-sensitization of highly-resistant clinical isolates due to severe PBP2a unfolding, in agreement with a number of publications that show the adjuvant properties of different statins to increase the bactericidal activity of antibiotics[29,87,110,111]. This work provides fundamental knowledge about the structural and functional importance of cellular membrane microdomains and shows that bacterial FMM spatially organize a sophisticated ATP-independent cellular process for proteome repair essential for bacterial survival during infections.

## Methods

This study complied with all biosafety and ethical regulations and was approved by the Institutional Review Board of the Spanish National Center of Biotechnology (CNB) and the Spanish Research Council (CSIC). Animal infection studies were reviewed by the Institutional Review Board of CSIC previous to being reviewed and approved by the Madrid Regional Government.

### Experimental models, strains, media and culture conditions

**Bacterial strains.** All bacterial strains used in this study are listed in Supplementary Dataset 7. The methicillin-sensitive *Staphylococcus aureus* strain Newman[112], the methicillin-resistant *S. aureus* (MRSA) strains USA300[113] and JE2[114], and the multi-drug resistant MRSA clinical isolates (obtained from the *Hospital Ramon y Cajal*, Madrid, Spain), were those used in experiments unless otherwise stated. For cloning purposes, *Escherichia coli* DH10b (Sigma) was used *E. coli* IM0BB served as the recipient strain[115]. For protein production, *E. coli* BL21 (DE3) Gold or Rosetta was used and BTH101 was used for bacterial two-hybrid (B2H) assays.

**Bacterial growth conditions.** *S. aureus strains* were cultured in TSB medium supplemented with erythromycin (2 µg/ml), kanamycin (10 µg/ml) or spectinomycin (50 µg/ml) where appropriate. *E. coli* strains were cultured in LB medium with ampicillin (100 µg/ml) when required. To avoid precipitation in an aqueous solution, ZA was prepared in dimethylsulfoxide (DMSO) stock solution and diluted 1:1 in methanol before further dilution in PBS, or in *S. aureus* cultures, to working concentrations. *E. coli* DH10b and BL21 (DE3) Gold/Rosetta were grown on Luria Bertani broth (LB) plates, in liquid LB medium or 2xTY medium with agitation at 37 °C. Ampicillin (100 µg/ml) or kanamycin (50 µg/ml) were added as required. The BTH101 strain for the B2H assays was grown at 30 °C.

**Construction of knock-out mutants.** ΔfloA, ΔnfeD and Δcrt mutants were generated using *a two-step recombination process*[116]. 700 bp upstream and downstream fragments were joined via long flanking homology PCR, and cloned into pMAD[116]. These were then transformed in *E. coli* IM08, the plasmid was propagated, purified, and used to

transform *S. aureus* via electroporation. After the successful integration of the entire plasmid into the genome, strains were grown at 42 °C, plated on TSB X-Gal (50 µg/ml), and white colonies screened for loss of the plasmid and the presence of the target construct by colony PCR. The resulting constructs were verified by DNA sequencing. The primers used are listed in Supplementary Dataset 8.

**Generation of labeled strains.** The pAmy and pLac plasmids were used for ectopic expression in *S. aureus*[117]. All the strains used bore the genetic constructs chromosomally integrated at the *amyE* or *lacA* neutral loci of *S. aureus*[116,117]. Sequence-validated plasmids were transformed in *S. aureus* by electroporation. The complemented strains harbored a copy of the gene of interest cloned in the pJL74 plasmid.

**Gene expression analysis by qRT-PCR.** *S. aureus* cultures were harvested and the cells were resuspended in lysis buffer (20 mM Tris, 10 mM EDTA) supplemented with lysostaphin (0.1 mg/ml) and incubated at 37 °C for 15 min. Total RNA was isolated using the RNeasy kit (Qiagen). RNA samples were treated with DNase I (New England Biolabs) to remove all DNA traces. RNA purity and quality were determined using Nanodrop technology (Thermo Scientific). DNA-free RNA samples were used for cDNA synthesis, employing Superscript III reverse transcriptase (Applied Biosystems) and random hexamer primers. Each PCR was run in triplicate using the Universal SYBR Green Master mix (Applied Biosystems). The designed primers were used to amplify a 120–200 bp DNA fragment. Relative amplification was calculated using the $2^{-\Delta\Delta CT}$ Livak Method[118]. The housekeeping *gyrB* rRNA gene was used as a reference.

**Cell fractionation and purification of the DRM and DSM fractions**
Pellets of *S. aureus* TSB cultures were harvested and resuspended in PBS buffer supplemented with 1 mM phenylmethylsulfonylfluoride (PMSF) and 5 µl DNaseI (2000 U/ml). For enzymatic lysis of the cells, 15 µl lysostaphin (1 mg/ml) was added and the cell suspensions incubated (15 min, 37 °C). These were then disrupted using three pulses of five minutes in a Genogrinder (SPEX). Unbroken cells and debris were removed by centrifugation ($10,000 \times g$, 20 min, 4 °C) and the supernatant ultracentrifuged ($100,000 \times g$, 1 h) to separate the membrane fraction. The pellet was dissolved (overnight, 4 °C) in 100–200 µl lysis and separation buffer (Sigma). For FMM isolation, the membrane fraction was processed using the CellLytic MEM protein extraction kit (Sigma)[26]. DRM and DSM fractions were separated according to the manufacturer's protocol. Samples were analyzed by SDS-PAGE. Cells were treated with the amine-reactive crosslinker dithiobis succinimidyl propionate (DSP) (1 mM) when required. Samples were incubated on ice for 2 h and the reaction then quenched by adding Tris-HCl pH 7.5 (10 mM) and incubating for 15 min on ice. Treated and untreated membranes were used immediately in further procedures or mixed with glycerol (10%) and flash-frozen in liquid nitrogen for storage at −80 °C.

**Separation of soluble and non-soluble proteins**
We used an already-described protocol[79] with some modifications. Cells from 10 ml cultures were collected, their membrane fraction purified and dissolved (overnight, 4 °C) in 1000 aqueous buffer (PBS buffer). 0,1% DDM was added to the buffer to allow the resuspension of the membrane fraction. Insoluble proteins were precipitated by centrifugation ($20.000 \times g$, 45 min, 4 °C) and quantified in relation to soluble proteins in all mutants tested.

**Immunodetection assays**
Unless differently stated in the results section, a standardized sample of 80 µg of total protein was resolve in 12% SDS-PAGE. Proteins were transferred to a PVDF membrane by semi-dry blotting (2 h), which was blocked with 5% skim milk and human serum 2% (1 h) and probed with specific antibodies. Proteins were detected using a chemiluminescent substrate kit (Thermo Scientific) and recorded with the ChemiDoc Image System (Bio-Rad).

**Protein production and purification**
Full-length FloA (1–329 aa) and variants were cloned in pET20b, and the protein overproduced in *E. coli* BL21 Rosetta (Novagen). Protein production was induced with 0.5 mM IPTG; cultures were grown overnight at 18 °C. Cells were harvested and resuspended in 50 mM Tris-HCl pH 8, NaCl 500 mM, 5% glycerol buffer containing 1 mM PMSF and protease inhibitors (Roche). Cell disruption was performed by sonication followed by centrifugation ($13,000 \times g$, 30 min, 4 °C) to discard debris. The extracts were then ultracentrifuged ($100,000 \times g$, 1 h, 4 °C) to separate membranes from the cytoplasmic fraction. The membrane fraction was resuspended in 50 mM Hepes pH 8, 300 mM NaCl, 0.5% DDM and incubated (1 h, 4 °C). Soluble membrane extracts were incubated for 1 h at 4 °C with 500 µl of Ni-NTA resin (Quiagen). Bound proteins were washed with buffer: 50 mM Tris-HCl pH 8, NaCl 500 mM, 5% glycerol, 0.05% DDM, 10–50 mM imidazole and eluted with 250 mM imidazole. No DDM was added to the Ni-NTA buffers to purify higher-order oligomers to maintain the natural lipid composition. For the purification of soluble variants (i.e., the PHB domain of FloA or the OBL of NfeD), the cytoplasmic fraction of the cell cultures was used; the membrane fraction was discarded.

To purify the FloA-NfeD complex for Cryo-EM studies, FloA and NfeD were coproduced in *E. coli* BL21 Rosetta (Novagen) using pETduet-1. This vector allowed the expression of His-tagged FloA and Flag-tagged NfeD in the same strain. Six-liter cultures were used for the purification process. A solubilized membrane fraction was obtained as described above. The suspension was then incubated with the Flag resin for 2 h at 4 °C. The resin was washed and proteins eluted from the resin twice with 300 µl Pierce-3x DYKDDDDK Peptide containing DDM (0.05%), and solubilized in a total volume of 1 ml. 100 µl of washed HisPurTM Cobalt Resin was then mixed with the eluted fraction and incubated overnight at 4 °C on the turning wheel. After washing, the protein complex was eluted from the resin with 200 µl elution buffer (250–500 mM imidazole).

NfeD and FloA variants were produced and purified using the pET29b vector (Novagen). This plasmid bears a C-terminal His-Tag sequence and a T7 promoter, allowing gene expression to be induced by IPTG. For overexpression, plasmids were transformed in *E. coli* BL21 (DE3) Gold (Novagen). Protein production was induced with 0.5 mM IPTG and cultures were grown for 5 h at 30 °C. The membrane fraction of 4 L cultures was resuspended in 10 ml of buffer (50 mM phosphate pH 8, 300 mM NaCl, 10 % glycerol), supplemented with DDM (1%) and solubilized in a turning wheel for 2 h at 4 °C. To eliminate insoluble material, the suspensions were cleared by ultracentrifugation ($100,000 \times g$, 30 min, 4 °C), and then mixed with 500 µl of Ni-NTA resin (Qiagen). Bound proteins were washed with 20× column volume wash buffer (50 mM phosphate pH 8, 300 mM NaCl, 10% glycerol, 0.05 % DDM, 50 mM imidazole) and eluted with 250–500 mM imidazole.

The two PBP2a variants, WT PBP2a (1–668 aa) and PBP2a$_{\Delta TMR}$ (23–668 aa), were overproduced and purified using pET20b and pET28b, respectively. The plasmid-encoded PelB signal sequence was cleaved by the membrane-bound signal peptidases. Cell pellets were resuspended in 50 mM Tris-HCl pH 8, NaCl 500 mM, 5% glycerol buffer containing 1 mM PMSF and protease inhibitors (Roche). Cell disruption was performed by sonication followed by centrifugation to eliminate the cell debris ($13,000 \times g$, 30 min, 4 °C). The cell extracts were ultracentrifuged ($100,000 \times g$, 1 h, 4 °C) to separate the membranes from the cytoplasmic fraction, and the membrane fraction then resuspended in 50 mM Hepes pH 8, 300 mM NaCl, 0.5% DDM and incubated for 1 h (4 °C). This was followed by ultracentrifugation ($100,000 \times g$,

1 h, 4 °C) to remove the insolubilized material. Membrane extracts were filtered (0.2 μm) and purified using the AKTA Pure system, employing 1 ml Ni-NTA pre-packed columns (GE Healthcare). These columns were then washed with buffer (50 mM Hepes, pH 8, 300 mM NaCl, 0.05% DDM) and the His-tagged proteins eluted with 250 mM imidazole-containing buffer. Protein purity was analyzed by SDS-PAGE. Proteins were buffer exchanged using PD-10 desalting columns equilibrated with 50 mM Hepes pH 8, NaCl 100 mM, 0.05% DDM, and the eluent loaded into a HiTrap SP HP ion exchange column (GE Healthcare). The attached proteins were eluted using a 100–1000 mM NaCl gradient. Fractions containing purified protein were flash-frozen in liquid nitrogen and stored at −80 °C.

## CLMS (cross-linking coupled to mass spectrometry)
Purified FloA and NfeD (0.1 mg/ml) were mixed 1:1 and disuccinimidyl suberate (DSS) added to the reaction mixture at a molar ratio of 1:50. These samples were then incubated at room temperature for 30 min, the reaction stopped by adding 0.1 M Tris pH 7.5, and the resulting mixture resolved by SDS-PAGE. The bands for the different oligomeric states were cut out and analyzed by mass spectrometry.

## Purification of STX
*S. aureus* culture pellets were resuspended in 10 ml methanol and incubated (30 min, 60 °C) with continuous agitation. Samples were centrifuged to remove cell debris and the pigment-containing supernatant was concentrated in a Speedvac (Thermo). STX were extracted with 2 ml ethyl acetate/1.7 M NaCl (1:1 v/v). After centrifugation, the ethyl acetate phase containing STX was recovered and the aqueous phase re-extracted with 0.5 ml ethyl acetate[41]. The extract was then dried *in vacuo*. Samples were resuspended in 1 ml of chloroform and resolved in a thin layer chromatography (TLC, Silica gel 60 Merck). The pigments were resolved using hexone:acetone (60:40 v/v) as mobile phase. The yellow pigmented band of STX was scratched from the gel and the powder resuspended in 1 ml of ethyl acetate/1.7 M NaCl (1:1 v/v) followed by centrifugation. The ethyl acetate phase containing STX was recovered and dried *in vacuo* to store at −20 °C. Carotenoid extracts were analyzed by HPLC using a HPLC Agilent 1100 with detector of Diode Array, a ProntoSil C30 stainless steel column and UV light detection at 450 nm. The compounds (50-μl injection sample) were separated with an acetone/water gradient (0 min, 90% acetone; 0–20 min, linear gradient to 100% acetone; 21–30 min, 100% acetone) at a flow rate of 1 ml/min. Mass spectroscopy was performed Bruker Maxis II ion-trap mass spectrometer equipped with an electrospray ionization source. The mass spectra was recorded in positive mode and external calibration of exact mass. The samples were dissolved in methanol and introduced into the equipment through an infusion syringe.

## Liposome-flotation binding assay
Solutions of STX, PE (Sigma) and PG (Sigma) in chloroform (0.4 mg) were dried under nitrogen flow (1 min) and kept under vacuum for 2 h. Multilamellar vesicles (MLV) were obtained by hydration in 100 μl 50 mM Tris-HCl, 150 mM KCl pH 7.5 buffer (final lipid concentration 4 mg/ml) followed by sonication (10-20 min or until the solution was transparent)[119]. MLV were kept at −20 °C. PHB variants (0.8 μg) were mixed with MLV and incubated for 15 min (room temperature). The PHB-lipid mixture was then diluted 1:1 with 60% sucrose (w/v in 50 mM Tris-HCl pH 7.5, 200 mM NaCl) to form the bottom layer of a sucrose gradient (200 μl), which was overlain with 250 μl 25% sucrose (w/v, same buffer), plus a top layer of 100 μl buffer (0% sucrose). Samples were centrifuged (270,000 × g, 1 h, 4 °C), and top fractions analyzed by immunoblotting to detect the PHB variants. The protein content of each fraction was precipitated with 10% trichloroacetic acid (TCA), dried, and resuspended in PBS prior to immunoblot analysis.

## Glycerol gradients
*S. aureus* cultures were harvested, resuspended in PBS in the presence of purified membrane fractions (200 μg), and solubilized with 0.5% DDM. They were then layered on a 10-30% linear glycerol gradient produced using a gradient maker (Biocomp), and centrifuged (100,000 × g, 16 h, 4 °C). Fractions (0.4 ml) were collected from the top of the gradient. Glycerol gradients were also used to separate the purified FloA oligomers as well as to isolate the FloA dimers bound to PBP2a. The gradients were prepared in 50 mM Tris, 250 mM NaCl, pH 7.5; no DDM was added to maintain the oligomerization of FloA and FloA-NfeD while allowing for PBP2a unfolding. For soluble protein variants, the glycerol gradients were prepared in 50 mM Tris, 250 mM NaCl, pH 7.5. For protein analysis, the protein content of each fraction was precipitated with 10% TCA. When required, 1 μl of Bocillin-fluorescently labeled β-lactam (Boc-FL 1.5 nM) was added to the sample and incubated for 30 min at 37 °C. Samples were analyzed by SDS-PAGE and the Boc-Fl signal detected using ChemiDoc Imaging System (Bio-rad).

## In vitro FloA-NfeD functional protein stabilization assay
Purified full-length PBP2a (2 μM) resuspended in purification buffer (50 mM Hepes pH = 8, 300 mM NaCl, 0.05% DDM) was incubated 15 min at different temperatures: RT, 37°, 42°, 55°. The reaction was run in parallel in the presence of FloA, FloA-NfeD or OBL (2 μM). All the samples were then centrifuged at 10,000 × g for 5 min and the supernatant collected. The remaining pellet was resuspended in and an equal volume of reaction buffer. Loading buffer was added to the samples, and these were analyzed on 12% SDS-PAGE gels (stained with Coomassie Brilliant Blue).

## Quantification of PBP2a activity in vitro by LC/MS
Purified PBP2a (2 μM) was incubated at 37 °C or 42 °C for 15 min with or without purified FloA (2 μM) and/or NfeD (2 μM) in the reaction buffer (50 mM HEPES, pH 6.5, 2.5 mM MgCl₂, 0.05 % DDM). After the heat treatment, *S. aureus* Gly5-Lipid II (40 μM) and PBP2$^{S398G}$ (2 μM) were added and the reaction mixture was incubated for 3 h at RT. *S. aureus* PBP2 variant containing an inactive transpeptidase domain (PBP2$^{S398G}$) was overexpressed and purified as previously described in ref. 85. To evaluate PBP2a refolding activity in vitro, purified PBP2a (2 μM) was pre-incubated at 42 °C for 5, 15, or 30 minutes. After this, samples were incubated with or without FloA (2 μM) for 15 minutes, followed by the addition of *S. aureus* Gly5-Lipid II (40 μM) and PBP2$^{S398G}$ (2 μM) to the reaction mixture and 3 h incubation at RT.

The enzymatic reaction was quenched at 95 °C for 5 min followed by 2 h treatment with mutanolysin (Sigma, 2 U) at 37 °C. Sodium borohydride (10 μL of 10 mg/mL solution) was used to reduce the resultant disaccharides for 30 min. The pH was then adjusted to 4 by adding 2 μL of 20% phosphoric acid. The reaction mixture was lyophilized, redissolved in 12 μL of MiliQ water and subjected to LC/MS analysis, conducted with Ultra-Performance Liquid Chromatography system interfaced with a Xevo G2/XS QTOF mass spectrometer (Waters). Chromatographic separation was achieved using an ACQUITY UPLC BEH C18 Column (2.1 mm × 150 mm, 1.7 μm pore size. Waters Corp.) heated at 45 °C. 0.1% formic acid in Milli-Q water (buffer A) 0.1% formic acid in acetonitrile (buffer B) were used as eluents. The gradient of buffer B was set as follows: 0–3 min 5%, 3–6 min 5–6.8%, 6–7.5 min 6.8–9%, 7.5–9 min 9–14%, 9–11 min 14–20%, 11–12 min hold at 20% with a flow rate of 0.175 ml/min; 12–12.10 min 20–90%, 12.1–13.5 min hold at 90%, 13.5–13.6 min 90–2%, 13.6–16 min hold at 2% with a flow rate of 0.3 mL/min; and then 16–18 min hold at 2% with a flow rate of 0.25 mL/min. The following ions were extracted from each chromatogram: monomer: 1253.5856 (M + 1); and dimer: 1209.0617 ((M + 2)/2). The QTOF-MS instrument was operated in positive ionization mode. The collision energy was set to scan between 6 eV and 15–40 eV. Mass spectra were acquired at a speed of 0.25 s/scan and the scans were in a

range of 100–2000 m/z. Data acquisition and processing were performed using UNIFI software package (Waters Corp.).

## Lipid II extraction and LC/MS analysis of delipidated Gly$_5$-Lipid II

Lipid II was extracted as previously described in ref. 120 with some modifications. A 1.5 L culture of *S. aureus* RN4220 was grown at 37 °C with shaking until OD$_{600}$ = 0.5–0.6. Moenomycin (0.6 μg/mL) was added to the culture to accumulate Lipid II; cultures were incubated for an additional 20 min before cells were harvested. The pellets were resuspended in 15 mL PBS (pH 7.4), 52.5 mL CHCl$_3$:MeOH (1:2) was added, and the mixture was stirred for 1 h at RT to allow cell lysis. The mixture was poured into two Teflon tubes and centrifuged at 4000 × *g* for 10 min at 4 °C. The supernatant containing the solubilized cellular contents was collected. The supernatants of two tubes were combined and poured into a flask containing 30 mL CHCl$_3$ and 22.5 mL PBS (pH 7.4). For an hour, the liquid was swirled rapidly to combine the layers thoroughly. The homogenized mixture was poured into three clean Teflon tubes and centrifuged at 4000 × *g* for 10 min at 4 °C. An interface fraction was revealed in each Teflon tube between the top aqueous and bottom organic layers. The aqueous layer was progressively removed to get to the interface fraction. The combined interface was dried on *Speedvac*. In the second extraction (to remove Park nucleotide), the combined dried interface was dissolved in a 15 mL organic mixture of 6 M pyridinium acetate: n-butanol (1:2) (6 M pyridinium acetate was prepared by mixing 51.5 mL glacial acetic acid with 48.5 mL pyridine), and washed with 15 mL of aqueous solvent (n-butanol saturated water) in a separatory funnel. The aqueous layer was extracted again with 10 mL organic solvent (1:2, 6 M pyridinium acetate: n-butanol) to maximize Lipid II extraction. The organic layers were combined and washed with aqueous solvent (n-butanol saturated water) three times (10 mL ×3) to remove the water-soluble Park nucleotide. On *Speedvac*, the pure organic layer was concentrated before being redissolved in DMSO.

To remove the Lipid II tail, the sample resuspended in DMSO was incubated with 800 μL of water and 100 μL of 0.1 M ammonium acetate pH 4.2. The mixture was boiled at 100 °C for 90 min, dried on *Speedvac*, resuspended in 50 μL of water, and centrifuged at 16,000 × *g* for 10 min to remove the precipitate. The pH of the supernatant was adjusted to pH 3–4 and the sample was subjected to LC/MS analysis.

## Size-exclusion chromatography

Purified PBP2a, FloA and NfeD variants were adjusted to concentrations of ~60 μM and separated by size-exclusion chromatography SEC in an Äkta high-performance liquid chromatography (HPLC) system. We used a Superdex 200 Increase 10/300 GL size-exclusion column (GE Healthcare) in the case of FloA variants and a Superdex 75 Increase 10/300 GL (GE Healthcare) in the case of NfeD variants. For each size-exclusion, 500 μl protein samples were loaded onto a column equilibrated with buffer (300 mM NaCl, 50 mM Tris-HCl pH 8.0, 10% glycerol, 0.02% DDM) and run at a constant flow rate of 0.4 ml/min. The purification of the apo FloA-NfeD dimer and bound to PBP2a for Cryo-EM examination was performed using using a Superdex 200 Increase 5/150 size-exclusion column (GE Healthcare). 10 μl protein samples were loaded onto a column equilibrated with buffer 200 mM NaCl, 50 mM Tris-HCl pH 8.0, 10% glycerol and run at a constant flow rate of 0.2 ml/min. Protein elution profiles were compared by UV absorbance (280 nm) of the chromatograms, and the graphs overlaid using PRISM software. A set of standard proteins used to calibrate the gel filtration column serve as theoretical MW markers (SigmaMWGF1000 and Cytiva LMW).

## Fluorescence microscopy

Cells from liquid cultures were washed in PBS and resuspended in 0.5 ml 4% paraformaldehyde (6 min, room temperature). Samples were washed twice and resuspended in 0.5 ml PBS. Images were then acquired with a Leica DMI6000B microscope equipped with a CRT6000 illumination system, an HCX PL APO oil immersion objective (100 × 1.47), and a DFC630FX color camera. Leica Application Suite Advanced Fluorescence Software was used for linear image processing. The YFP signal was then sought (excitation filter 489 nm, emission filter 508 nm). Excitation times were 567 ms. Transmitted light images were taken with 55 ms excitation times.

Microscopy images were analyzed and processed using Fiji software[121]. To quantify the number of fluorescent membrane foci per cell, maximum intensity projection (MIP) images were generated using Fiji, in which the pixels are represented with the highest signal intensity in a final image as a direct analysis method. We used the plugin Trackmate[122] to semi-automatically segment membrane foci from a 2D image. Trackmate detects foci according to the input parameters that were set by the Laplacian of Gaussians (LoG) detector. The LoG detector applies a plain LoG segmentation on the image. Calculations are made in the Fourier space, for spot sizes between 5 and 20 pixels in diameter. Estimated blob diameter, 0.3 μm; threshold, 200; perform subpixel localization, no median filter, and no filter on object identification.

## Bacterial two-hybrid analysis

Coding sequences were cloned into bacterial two-hybrid expression vectors (EuroMedex) to generate N- and C-terminal fusions with the catalytic domains T25 and T18 of *Bordetella pertussis* adenylate cyclase. Pairwise combinations of plasmids were used to co-transform *E. coli* BTH101, which harbors a *lacZ* gene under the control of a cAMP-inducible promoter. After their interaction, the T25 and T18 catalytic domains form an active enzyme, leading to cAMP production and reporter expression. Plates were incubated (48 h, 30 °C). pKT25-zip and pUT18C-zip, and pKT25 and pUT18C, served as positive and negative controls, respectively. β-galactosidase levels were recorded in Miller units.

## Pull-down assay

Pull-down assays were performed in polypropylene Qiagen columns with 1 ml bed of Ni-NTA resin (Qiagen). Columns were loaded with supernatant and samples and incubated for ON (4 °C) to allow the proteins to bind to the resin. Samples were centrifuged (1000 × *g*, 2 min) after each step and the supernatant was removed. The columns were washed by gravity-flow with low-imidazole Hepes buffer (50 mM Hepes, 100 mM NaCl and PMSF, 10–20 mM Imidazole, pH 7.5) and loaded with 700 μl of solubilized membrane fractions of *S. aureus* cells producing NfeD or the NfeD$_{SSDL}$ variant. To remove unbound and non-specific proteins, the resin was washed for 3 h on a turning wheel with low-imidazole Hepes buffer. For elution, 1 ml buffer containing 100 mM imidazole was added and the protein content of this fraction was TCA precipitated and processed for immunoblot analysis.

## Blue-native PAGE (BN-PAGE)

Cultures were grown in TSB medium (24 h, 200 rpm), the cells harvested, and pellets dissolved in PBS buffer containing 1 mM PMSF and complete protease inhibitors (Roche). Samples were crosslinked with 0.5 mM DSP before cell lysis and fractionation. The membrane fraction (~80 μg) was solubilized in 1× Native PAGE sample buffer (Invitrogen) with 0.5% DDM (overnight, 4 °C). The solubilized membranes were then mixed with Coomassie dye G-250 and loaded on a native gel with a 3–12% polyacrylamide gradient (Invitrogen). BN-PAGE was performed as previously described[123]. BN-PAGE uses Coomassie G-250 to confer a negative charge to proteins, allowing the resolution of oligomeric complexes according to their native state. Native gels were incubated in PBS + SDS (1%) for 15 min before transfer to PVDF membranes. These membranes were then fixed with 8% acetic acid for 15 min, air-dried, and rewetted with methanol prior to blocking with milk and incubation with specific polyclonal antibodies.

## Negative-staining electron microscopy: sample preparation and imaging

Samples were initially screened by negative-staining EM before Cryo-EM examination. Purified protein (5 μl at 0.02 mg/ml) was deposited on a copper grid covered with a carbon layer and incubated for 2 min at room temperature. The excess sample was blotted with Whatman filter paper. Then, the grid was placed the drop of a 2% uranyl acetate drop was then placed on the grid placed and incubated for 2 min at room temperature. The excess was blotted off and the grid left to dry before image capture using a JEM1400 Flash (Jeol) electron microscope operating at 120 kV and equipped with a CMOS Oneview (Gatan). Upon image collection (FloA 89; FloA-NfeD 112; 2xFloA-NfeD 102 and 2xFloA-NfeD+PBP2a 120 images), CTF was corrected using CisTEM-CTFfind4 software[124] within the Scipion platform. For each sample, a set of 10 images was used for manual picking in Xmipp followed by Autopicking also in Xmipp. Selected particles were extracted and subjected to 2 rounds of 2D classification in cryoSparc[125] (in Scipion platform). 3D initial model was created using Ransac program in Scipion, the best model was chosen for homogenous refinement in CryoSparc.

## Cryo-electron microscopy: sample preparation and imaging

Extensive screening of the type of grids and vitrification conditions was performed for the cryo-EM sample preparation was performed using a 200 kV TALOS Artica cryo-electron microscope equipped with an autoloader and a Falcon III direct electron detector (CNB-CSIC. Madrid, Spain). The types of grids tested were Quantifoil 2/2 300 mesh +2 nm carbon; Quantifoil 1.2/1.3 300 mesh; Quantifoil Au 1.2/1.3 300 mesh; Quantifoil Au 0.6/1 300 mesh and Ultrafoil Au 2/2 300 mesh. For all grids, we tested for 1 min, 30-sec, or 20-sec glow discharge and no glow or positive glow discharges (1 min). We used FEI Thermo Fisher Vitrobot IV for the vitrification conditions and tested a blot time range of 4–7 sec and −3 to −5 of blot force. The conditions selected for data collection involved 3 μl of the sample (0.02 mg/ml) placed in Ultrafoil Au 2/2 300 mesh grids and prepared by glow discharging (1 min, 25 mA). Samples were incubated for 10-30 s and plunge-frozen using a FEI Vitrobot IV device (blot force −4, blot time 4–5 s) with 100% humidity and 22° temperature. For Data collection, we used two different cryo-EM facilities equipped with a 300 kV Titan Krios cryo-electron microscope; the Basque Resource for Electron Microscopy (BREM) (Bilbao, Spain) and the Umëa Center for Electron Microscopy (UCEM) (Umëa, Sweden). For cryo-EM of monomers (FloA and FloA-NfeD), 9653 movies were recorded at a pixel size of 0.58 Å/px on Titan Krios 300 keV microscope, Falcon 4i camera using AFIIS method. The defocus range used: −1.2 to −2.4 μm with 0.3 step size with a total dose of 60 e/A$^2$. The defocus range used: −1.5 to −3.0 μm with 0.3 step size with the total dose of 60 e/A$^2$. To generate the map of the 2xFloA-NfeD, a total of 19615 movies were recorded at a pixel size of 0.5054 Å/px on Titan Krios 300 keV microscope, K3 camera. The defocus range used: −1.2 to −2.6 μm with 0.3 step size with the total dose of 47.5 e/A$^2$. The map of the 2xFloA-NfeD + Pbp2a complex a total of 12340 movies were recorded at a pixel size of 0.6462 Å/px on Titan Krios 300 keV microscope, K3 camera. The defocus range used: −1.2 to −2.6 μm with 0.3 step size with a total dose of 48 e/A$^2$.

## Image processing and map interpretation

All images were processed using the Scipion platform[126]. We used ColabFold v1.5.2. (AlphaFold2) for the structure prediction of FloA and NfeD. We used AlphaFold2 Multimer to predict the structures of FloA and FloA-NfeD dimers.

For the FloA-NfeD monomer (1xFloA-NfeD), the motion correction was performed using Patch Motion Correction from CryoSPARC software. The dose-weighted micrographs were imported to Scipion and CTF corrected using the Cistem-ctffind4 program. Xmipp CTF consensus protocol was used to select micrographs with an astigmatism threshold of less than 20% and the resolution threshold of 5 Å.

A set of 19 micrographs was used for Manual picking in Xmipp, followed by Xmipp Autopicking using a set of 8387 micrographs. Selected particles were subjected to 3 rounds of 2D classification followed by ab initio reconstruction in CryoSPARC (implemented within Scipion platform) with two classes (1435678 particles). Particles from one class (238299) were pruned by ab initio reconstruction with two classes. One class (40774 particles) was refined using Local refinement protocol from CryoSPARC using a soft mask covering the entire structure. The samples contained small membrane proteins (37 and 72 kDa), so the image SNR was relatively low. Therefore, we tested different refinement protocols in CryoSPARC software, such as 3D homogenous refinement or 3D non-uniform refinement. The final output was obtained using Local refinement by CryoSPARC. AlphaFold2 prediction was flexibly fitted using iMODFit[127] software. Resolved and fitted region of FloA: aa 127–329, NfeD: aa 173–235.

The structure of the FloA monomer (1xFloA) was previously obtained from data collected in TALOS 200 kV microscope using the same grid. The reprojections of the obtained EM map were used as a reference for autopicking (Gautomatch) using a set of dose-weighted micrographs (8387 micrographs). From this, 986707 particles were picked for one round of 2d classification using CryoSPARC. A set of 478,555 particles was cleaned by ab initio reconstruction with four classes. One class (118,042 particles) was refined using non-uniform refinement (a final map was obtained at 8.0 Å of resolution) followed by local refinement with soft static mask. Fitting was performed using iMODFIT flexible fitting software. AlphaFold2 prediction model was flexibly fitted into the EM map, resolved region: aa 51- 329. Volumes were visualized and the animations were prepared in UCSF Chimera software[128]. Cryo-EM 3D maps were deposited in EMDB with the codes EMD-1722 (1XFloA-NfeD) and EMD-17217 (1XFloA).

For the FloA-NfeD dimer (2xFloA-NfeD), the pre-processing of the movies followed the previously established working pipeline (Patch motion correction in CryoSPARC; CTF correction using cistem-ctffind4; Xmipp CTF consensus [threshold: 10% astigmatism, <6 Å resolution]). A final set of 14960 micrographs was obtained. A set of 15 micrographs was used for manual picking in Xmipp followed by Xmipp Autopicking using a set of 14960 micrographs. Selected particles (3706679 particles) were subjected to one round of 2D classification followed by ab initio reconstruction using CryoSPARC with four classes (483832 particles). Particles from one class (137574) were pruned by ab initio reconstruction with three classes. One class containing 48648 particles was refined using non-uniform refinement followed by local refinement from CryoSPARC using soft mask covering the entire structure. The final round of Local refinement included C2 symmetry imposed, as suggested by the results obtained by SEC, Blue-Native and negative-staining EM analysis. A cryo-EM map was obtained at 8.8 Å of resolution. AlphaFold2 prediction was flexibly fitted into the map using iMODfit software. The fitted resolved region covered: aa 127–329 for FloA and aa 173–277 for NfeD (OB-fold domain). Cryo-EM 3D maps were deposited in EMDB with the code EMD-17231.

To obtain the structure of the FloA-NfeD dimer loaded with PBP2a (2xFloA-NfeD+1PBP2a), movie pre-processing was performed according to the established working pipeline (Patch motion correction in CryoSPARC; CTF correction using cistem-ctffind4; Xmipp CTF consensus [threshold: 10% astigmatism, <6 Å resolution]). From 7840 micrographs, 15 images were selected for manual Picking in Xmipp followed by Autopicking. 2,515,827 particles were finally picked, extracted and subjected to two rounds of 2D classification. The resulting 1,310,395 particles were subjected to ab initio reconstruction in CryoSPARC with four classes. Particles from one class (324,314 particles) were pruned with two rounds of ab initio reconstruction with two classes. One class (52403 particles) was refined using non-uniform refinement followed by local refinement in CryoSPARC using soft, static global mask. The map revealed a low-detailed density in the central volume attributable to unfolded Pbp2a and more structural

details assigned to the FloA-NfeD dimer. Therefore, we created a mask subtracting the Pbp2a signal. The particles were refined against the entire volume using a mask covering the FloA-NfeD dimer region and applying C2 symmetry during Local Refinement. Final map was obtained at 8.1 Å of global resolution. AlphaFold2 prediction was flexibly fitted into the map using iMODfit software. The fitted resolved region covered: aa 127–278 for FloA and aa 173–277 for NfeD (OB-fold domain). Unfolded Pbp2a EM density was segmented in Chimera. Cryo-EM 3D maps were deposited in EMDB with the code EMD-17233.

## MS-based proteomics of DRM and DSM

Protein samples were reduced in buffer with 50 mM DTT (5 min, 95 °C), alkylated with 120 mM iodoacetamide, precipitated with acetone, and dissolved in 0.5% sodium deoxycholate. Digests were performed with 0.5 µg LysC and 0.5 µg trypsin. SDC was removed by ethyl acetate extraction, desalted with C18 stage tips, and dissolved in 2% acetonitrile. NanoLC-MS/MS analyses were performed using an Orbitrap Fusion instrument equipped with an EASY-Spray ion source coupled to an EASY-nLC 1000 device (Thermo). Peptides were loaded on a trapping column (2 cm × 75 µm ID, PepMap C18, 3 µm particles, 100 Å pore size) and separated on an EASY-Spray column (25 cm × 75 µm ID, PepMap C18, 2 µm particles, 100 Å pore size) using a 120 min linear gradient (3–32%) of acetonitrile and 0.1% formic acid. MS scans were acquired in the Orbitrap analyzer. MaxQuant software was used for processing MS raw data files, database searches and quantification[129] against the UniProt *S. aureus* database. MS peak intensities were used for protein quantification[130]. Proteins with fewer than two identified razor/unique peptides were dismissed. The raw mass spectrometry data was deposited in the PRIDE repository[131] under the dataset identifier PXD006546.

## MS-based proteomics of soluble and insoluble protein fractions

Cultures were grown in a chemically defined medium[132] at 37 °C with 200 rpm until OD600 reached 3.0. Chloramphenicol 10 µg/ml was added to the cultures to stop protein synthesis, followed by incubation at 37 °C with 200 rpm for 15 minutes. Cultures were incubated at 55 °C for 30 min to allow protein unfolding by heat shock treatment, followed by incubation at 37 °C with 200 rpm for 30 min to allow protein refolding. Samples were taken after chloramphenicol addition, after the heat shock and after incubation recovery. Cells were harvested and their membrane fraction purified and dissolved (1 h, 4 °C) in an aqueous buffer (PBS buffer). 0,1% DDM was added to the buffer to allow the resuspension of the membrane fraction. Insoluble proteins were precipitated by centrifugation (20.000 × g, 45 min, 4 °C) and quantified in relation to soluble proteins in all mutants tested. 2 µg of soluble or insoluble samples were denatured with a final concentration of 2% SDS and 20 mM TCEP prior to being digested with a modified sp3 protocol[133], as previously described[134]. Briefly, a bead suspension (10 µg of beads, Sera-Mag Speed Beads) in 10 µl 15% formic acid and 30 µl ethanol was added to the samples and after incubation for 15 min at room temperature with shaking, beads were washed four times with 70% ethanol. Proteins were then digested overnight by adding 40 µl of digest solution (5 mM chloroacetamide, 1.25 mM TCEP, and 200 ng trypsin in 100 mM HEPES pH 8.5). After elution from the beads, peptides were dried under vacuum and labeled with TMT6plex (Thermo Fisher Scientific). After pooling, samples were desalted with solid-phase extraction using a Waters OASIS HLB µElution Plate (30 µm). Finally, samples were fractionated into 48 fractions on a reversed-phase C18 system under high pH conditions, pooling every sixth fraction together. Samples were analyzed by LC-MS/MS using a data-dependent acquisition strategy on a Thermo Fisher Scientific Vanquish Neo LC coupled with a Thermo Fisher Scientific Orbitrap Exploris 480. Raw files were processed with MSFragger[135] against a *S. aureus* FASTA database downloaded from Uniprot (UP000001939) using standard settings for TMT. Data were normalized using vsn[136] and statistical

significance was determined using limma[137]. The mass spectrometry proteomics data have been deposited to the ProteomeXchange Consortium via the PRIDE partner repository with the dataset identifier PXD041057.

## In vitro macrophage infection assays

Human monocyte THP1 cells (ATCC, TIB-202) were cultured in RPMI 1640 supplemented with 10% fetal bovine serum. Cells were maintained at 37 °C in a humidified atmosphere with 5% CO2. THP1 cells tested negative for mycoplasma contamination. THP1 cells were plated 72 h before infection at $6 \times 10^3$ CFU per well in black, clear-bottom 384-well plates (Greiner, 781090) for fluorescence microscopy-based infection assay and at a density of $2 \times 10^4$ CFU per well in 24-well plates for CFU assay. THP1 cells were differentiated into macrophage-like cells by treatment with 50 ng/ml phorbol 12-myristate 13-acetate, at the time of plating. Overnight bacterial cultures were diluted 1:100 in TSB and grown aerobically at 37 °C with shaking until OD600 0.4. Bacteria were harvested by centrifugation ($12,000 \times g$, 2 min) and resuspended in complete medium of the mammalian cells. Infections were performed at MOI of 25. Following the addition of the bacteria, cells were centrifuged at $500 \times g$, at room temperature for 10 min, and incubated at 37 °C in a 5% CO2 humidified atmosphere for 50 min. Extracellular bacteria were killed by replacing the medium with fresh medium containing 5 µg/ml lysostaphin (Ambi) and 100 µg/ml gentamicin. Hydrogen peroxide ($H_2O_2$, 1 mM) was added with the antibiotics. Medium supplemented with antibiotics and $H_2O_2$ was maintained until collection (1.5 hpi). To quantify intracellular bacterial replication by CFU assays, cells were washed three times with PBS and lysed in PBS containing 0.1% Triton X-100. The lysates were then serially diluted in PBS and plated on TSB agar plates.

For the fluorescence microscopy-based infection assay, cells were rinsed with PBS and fixed with 4% paraformaldehyde for 15 min at room temperature. After 30 min of permeabilization with 0.5 % Triton X-100 in PBS, S. aureus was labeled with 0.25 µg/ml BODIPY-FL vancomycin (Invitrogen, V34850) for 2 h at room temperature. Cells were washed, and nuclei were counterstained with Hoechst 33342 for 15 min at room temperature. Image acquisition was performed using an Operetta automated high-content screening fluorescence microscope (PerkinElmer), at ×20 magnification, with a total of 9 images acquired per well. Image analysis to quantify *S. aureus* intracellular load, bacterial replication, and host cell viability, was performed using custom workflows implemented in Columbus image analysis software (PerkinElmer), as previously described[138].

## Galleria infection studies

For infections of wax moth (*Galleria mellonella*) presented in Fig. 1D, larvae were obtained from DNATecosistemas (Spain). Larvae were injected with $10^6$ S. aureus CFUs in a MgSO$_4$ 10 mM solution into the last left proleg of each caterpillar and incubated at 37 °C for 48 h. Three independent experiments were performed using 15 larvae per tested strain and experiment. The control group was injected with 10 mM MgSO$_4$. Animal survival was monitored every 12 h. Larvae were considered dead when they did not react anymore to gentle tapping.

## Mouse infection studies

All animal studies were approved by the Madrid Regional Government, Spain (license no. 55.2-DMS-2532-2-57) and performed in strict accordance with the guidelines for animal care and experimentation according to Spanish law and EU directive 2010/63/EU. Female BALB/c mice (8-10 weeks old, body weight 16–19 g) (Charles River Laboratories) were housed in polypropylene cages under standardized lighting conditions and had *ad libitum* access to food and water.

*S. aureus* strains were cultured on TSB medium (18 h, 37 °C). Cells were collected, washed three times with PBS and diluted to OD$_{600 \, nm}$ = 0.05. Viable cell counts were determined by plating

inoculum dilutions on TSB agar plates. In the sepsis experiments showed in Fig. 1D, cohorts of 10 mice were infected with 150 µl of *S. aureus* cultures ($3 \times 10^7$ cells) via intravenous injection. The infection was allowed to progress for two days. Mice were sacrificed and the lungs were collected aseptically, homogenized, and plated on mannitol-agar for CFU counts. For the pneumonia infection model that is shown in Fig. 6F, cohorts of 10 mice were infected with 100 µl of *S. aureus* cultures ($3 \times 10^7$ cells) via nasal instillation. A single dose of oxacillin (200 mg/kg) alone or combined with ZA (50 mg/Kg or 20 mg/Kg) was injected intraperitoneally 30 min after infection. Infections were allowed to progress for 24 h. The mice were then sacrificed and the lungs were collected aseptically, homogenized, and plated on mannitol-agar plates for CFU counting.

### Statistical analysis

Graphs represent data from three independent biological replicates. Data are shown as mean ± SEM. The parametric unpaired two-tailed Student t test with Welch's correction, and the non-parametric unpaired Mann-Whitney test, were used to detect differences between pairs of groups. For experiments with three or more groups, the parametric one-way ANOVA test was used. *Post hoc* analysis involved the Tukey test, Dunnett's test or Dunn's test as required. Significance was set at $p < 0.05$. All calculations were performed using Sigma-Plot software (Systac Software).

### Reporting summary

Further information on research design is available in the Nature Portfolio Reporting Summary linked to this article.

### Data availability

For proteomic analyses, the raw mass spectrometry data were deposited in the PRIDE repository under the dataset identifiers: PXD006546: https://www.ebi.ac.uk/pride/archive?keyword=PXD006546 PXD041057: https://www.ebi.ac.uk/pride/archive?keyword=PXD006546 The cryo-EM 3D maps have been deposited in EMDB: 1XFloA EMD-17217: https://www.ebi.ac.uk/emdb/EMD-17217 1XFloA-NfeD EMD-17222: https://www.ebi.ac.uk/emdb/EMD-17222 2XFloA-NfeD EMD-17231: https://www.ebi.ac.uk/emdb/EMD-17231 2XFloA-NfeD+PBP2a EMD-17233: https://www.ebi.ac.uk/emdb/EMD-17233 A Source Data File including uncropped gel images are provided with this manuscript and they are also deposited in Figshare https://doi.org/10.6084/m9.figshare.25046069 Source data are provided with this paper.

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

## Acknowledgements
M.U. and M.K. received a MSCA fellowship from EU. I.R.L. is a recipient of PDBEB PhD fellowship. We thank A. Burton for editorial assistance. This work was funded by the Swedish Research Council (FC and DL), the Spanish Ministry of Science and Innovation and Fundación La Caixa (D.L.) and the Portuguese Foundation for Science and Technology (A.E.). We thank the BREM (Basque Resource for Electron Microscopy) and Umeå University cryo-EM facilities as well as Instruct-ERIC, Instruct Image Processing Center (I2PC) and IFCA for computer resources and cryo-EM support. Finally, we thank Daniel Kahne (Harvard University) for sharing the PBP2[S398G] variant and Tiago Costa (Imperial College, UK) and Alberto Marina (IBV-CSIC), for comments and suggestions in Cryo-EM imaging.

## Author contributions
Conceptualization, D.L.; methodology, M.U., A.E., M.L.B., R.M., A.I.R., A.M., F.C., and D.L.; investigation, M.U., L.K., G.T., I.P., I.L., M.K., J.G.F., R.M., M.L.B., A.I.R., A.M., F.C., and D.L.; writing—original draft, D.L.—review & editing, M.U., L.K., G.T., I.P., I.L., M.K., J.G.F., R.M., M.L.B., A.E., A.M., F.C.m and A.I.R.; Funding acquisition, resources & supervision, R.C., A.E., A.M., F.C., and D.L.

## Competing interests
The authors declare no competing interest
