## [Peer Review File · Nature Communications]

Flotillin-Mediated Stabilization of Unfolded Proteins in Bacterial Membrane MicrodomainsREVIEWER COMMENTS

Reviewer #1 (Remarks to the Author):

I am not an expert in lipid rafts or membrane protein experiments, mass spec proteomics experiments, and although I know a few things about cryo-em, I am not enough of an expert to evaluate the way the low-resolution data was interpreted. These are expertises that will be required in addition to mine for a complete review.

Major comments:

Fig 2Cii- the interpretation here is not very clear as it also looks like it doesn't bind in the control lane either.

On the BLI experiments, it appears only a single concentration was used. For BLI, multiple concentrations must be used to determine the k_{on} and K_d quantitatively. If not doing multiple concentrations, the wording must be considerably softened and not reported quantitatively.

The results have an issue in the protein refolding conclusions. The results are largely on solubility vs insolubility, not folded vs unfolded. This is an assumption that needs to be tested to claim this, as solubility and folding are not necessarily equivalent for proteins. Also, the suggestion that there is a direct effect on folding would need to be tested as well to be claimed; it is quite possible that there is an effect under stress to hold proteins to keep them soluble, but this could be a pure holdase mechanism. Even if there isn't a direct effect on folding, and the effect is that of a holdase and on solubility alone, that's still interesting. The transpeptidase activity assays are not sufficient in this case because they are stabilizing a binding partner in their folded state, as opposed to refolding the protein from an unfolded state. Many evolved binding partners will do this. The one piece of data suggesting a recovery of folding would be Fig 5C, which looks like it might be promising, but as far as I can tell, it was only performed once and the amount present is quite small; what is the error and is this statistically significant? Even if this is the case, this is still not enough to claim that there is a direct effect on folding on its own, as it could still be holdase activity with this level of data.

One specific thing for Fig 4A: show the total, soluble, and insoluble. At present, it's hard to tell where the proteins really go.

In 4B- the conclusions are overstated here, there are other possibilities.

In 4C- since there's more in the insoluble fraction with the mutants, there should be less in the soluble fraction if they were loaded correctly, but this does not appear to be the case.

In 5D- there's two issues. First, doing a 1:1 folded unfolded ratio could really not work as expected, since aggregation could continue. Second, the bands on the right of the gel are much more faint than on the left, such that I don't think you would see any of the substrate bound by eye on the right side gel fractions anyway even if it was there.

However, showing a direct role in folding definitively is not required to make a worthy paper- the collection of results they have is suggestive of a role in proteostasis, which is interesting.

The authors could remove results that have not been repeated and/or aren't statistically significant along these lines, tone down the wording on folding vs solubility, and could still have enough to publish here.

Minor points:

Fig 1A- a color problem, one of them should be gray.

For the experiments in Fig 1B and similar, couldn't find a materials and methods section, please add. For Fig 1D- can't find anything in the methods explaining how these were done as opposed to those in Fig 6.

Fig 1C- typo in the iii label. Also, some parts aren't labeled.

Fig 1D- could you add a bit more summary sentences for the multipart panels? For this one in particular.

Fig 2Di description in the text is quite unclear, not sure of the interpretation of the data by the authors here. The word remain is ambiguous.

For 3Ciii- couldn't find the materials and methods section for the immunodetection, please add.

4A: the total protein is total from the cell or total of membrane extracts?

43i: unit issue on the y axis- looks like infinity grams/ml

4E: there's a mislabel of ii in the caption as to the diagram.

Line 62: Pet peeve- nothing can be experimentally proven; this wording needs to be revised.

Reviewer #2 (Remarks to the Author):

In this manuscript, the authors provide compelling evidence for the critical role of flotillin membrane microdomains (FMM) in methicillin-resistant *Staphylococcus aureus* (MRSA) in confining and stabilizing unfolded proteins due to cellular stress. Using Cryo-EM analysis, they demonstrate that the PHB domain-containing proteins FloA and NfeD form clamp-shaped oligomers, creating a structure that holds accumulating unfolded proteins, thereby stabilizing them and aiding in their correct folding. Crucially, this process occurs without imposing a direct energy cost on the cell. The authors emphasize the significance of this mechanism for the survival of ATP-depleted bacteria and its impact on MRSA pathogenesis. The manuscript is well-written and illustrated. I have only a few minor concerns.

1. Figure 1A: The bars for bacterial survival of the Δ nfeD strain should be filled in gray.

2. Lines 122-123: The authors demonstrate that both H₂O₂-stressed cells and untreated cells collected at the stationary growth phase exhibit an increased number of foci compared to untreated, exponentially growing cells. Is this difference attributable to an up-regulation of

the transcription and/or translation of FloA-NfeD? Furthermore, are there regulatory motifs identified in the upstream region of the operon?

3. Lines 156-163: The authors propose that aromatic residues in the sterol-sensing domain (SSD)-like region (SSDL) are accountable for NfeD localization at FMM. However, MRSA cells do not have sterols in their cytosolic membranes. Which lipid species are responsible for NfeD recruitment at FMM? Has the binding of staphyloxanthin by NfeD been tested?

4. Expanding on the aforementioned concern: Since NfeD specifically interacts and co-localizes with FloA (as demonstrated in Fig. 2 and detailed in lines 176-181), and FloA is known to localize in discrete foci across the cell membrane (as mentioned in lines 121-122 and depicted in Fig. 1B), and considering that this interaction is independent of the SSDL motif, I am intrigued by the uniform distribution of the NfeDSSDL-YFP construct (with an altered SSDL motif) in MRSA membranes, as observed in Fig. 2Bii. How could this be explained?

5. Lines 274-277: the author mentioned "As FMM are highly hydrophobic membrane regions, they may provide a stabilizing environment to unfolded proteins that show their hydrophobic side-chains exposed to the solvent and contribute to isolating unfolded proteins from the rest of the cell to prevent unspecific interactions" Are FMM regions more hydrophobic than other parts of the membrane? Sterols, terpenoids, and carotenoid lipid species contribute to membrane order and reduce fluidity, but as far as I understand, they do not increase hydrophobicity. The authors should provide references or further explanation to support this point.

Reviewer #3 (Remarks to the Author):

Review comments

Authors describes in this manuscript flotillin (FloA) and nfeD that constitute functional membrane microdomains (FMM) from *Staphylococcus aureus* (MRSA) and that using Cryo-EM studies, flotillins forms a clamp-shaped oligomer and also holds the accumulating unfolded protein, indicating that this FMM system contributes to MRSA viability during infection, due to these proteins stabilize unfolded proteins and favor their correct folding under ATP-independent pathway.

Many readers and I will be interested in these studies. The function of FMM scaffold protein flotillin have been unknown, but authors have first clarified their function.

Major points:

I think this article has large problem. In this manuscript, authors showed several cryo-EM studies on 1xFloA, 1xFloA-NfeD, 2xFloA-NfeD, 2xFloA-NfeD and PBP2a, but their resolution are $\sim 9 \text{ \AA}$, too low. In such a resolution, authors cannot describe precise molecular shapes. Authors emphasize the word "Cryo-EM" in the abstract, but I do not recommend it. In the abstract, authors should present other enormous studies such as Co-IP, CLMS, BLI, B2H, fluorescence microscopy and so on, rather than Cryo-EM. Even studies other than Cryo-EM are valuable for this manuscript. Or authors should emphasize that this cryo-EM studies are challenging to resolve small proteins (below 100 kDa) (which are described in

Lines 204-208).

This article presented much information about FloA, NdeD, and FMM to stabilize unfolded proteins. That is good, but I have some difficulties to read this manuscript, because too much Figures, Extended Figures, and supplements are present, and one figure contains complicated data. I want this manuscript more compact and readable to catch the importance as much as possible.

Minor points:

Line 111: For the description “where is known to colocalize with flotillin (FloA)^{36,51}”, ref 36 reported that FloT colocalize with FloA. Authors should explain NfeD and FloT in more precisely.

Line 141: “one Nt transmembrane segment”, please spell out Nt at first.

Lines 195-196: “NfeD dimerization involves an OBL-OBL interaction”, please indicate why you can describe it using SEC analysis on Figure 2G, in which OBL shows one peak. I think this peak corresponds to a dimer. Is it correct ?

Lines 243-244: This sentence are referred to Fig 3Biii and Extended Data Fig. 5D. Please explain more precisely, which peaks in Extended Data Fig. 5Dii correspond to the peptides in Diii. I cannot find data.

Line 340: “hydrophobic interaction”, I do not understand why the recruitment of unfolded proteins at FMM is mainly driven by hydrophobic interaction. Please indicate.

Extended data, Figure 3 Di: Arrowheads corresponding to FloA and NfeD show different points. Please correct.

Extended data, Figure 7 D: The legend i may correspond to ii, and the legend ii may correspond to to iii. There are no legends of i.

Extended data, Figure 8: Legends of D and E may be opposite.

REVIEWER COMMENTS

Reviewer #1 (Remarks to the Author):

I am not an expert in lipid rafts or membrane protein experiments, mass spec proteomics experiments, and although I know a few things about cryo-em, I am not enough of an expert to evaluate the way the low-resolution data was interpreted. These are expertises that will be required in addition to mine for a complete review.

Major comments:

Fig 2Cii- the interpretation here is not very clear as it also looks like it doesn't bind in the control lane either.

We have extended the description of this panel (lines 172-177) and replaced the immunodetection assay with another one in which the PHB binding to the lipids shows a clearer signal.

On the BLI experiments, it appears only a single concentration was used. For BLI, multiple concentrations must be used to determine the k_{on} and K_d quantitatively. If not doing multiple concentrations, the wording must be considerably softened and not reported quantitatively.

We removed BLI experiments in the revised manuscript to align with the suggestions of this reviewer; of note, we previously reported BLI assays in Koch et al., 2017.

The results have an issue in the protein refolding conclusions. The results are largely on solubility vs insolubility, not folded vs unfolded. This is an assumption that needs to be tested to claim this, as solubility and folding are not necessarily equivalent for proteins. Also, the suggestion that there is a direct effect on folding would need to be tested as well to be claimed; it is quite possible that there is an effect under stress to hold proteins to keep them soluble, but this could be a pure holdase mechanism. Even if there isn't a direct effect on folding, and the effect is that of a holdase and on solubility alone, that's still interesting. The transpeptidase activity assays are not sufficient in this case because they are stabilizing a binding partner in their folded state, as opposed to refolding the protein from an unfolded state. Many evolved binding partners will do this. The one piece of data suggesting a recovery of folding would be Fig 5C, which looks like it might be promising, but as far as I can tell, it was only performed once and the amount present is quite small; what is the error and is this statistically significant? Even if this is the case, this is still not enough to claim that there is a direct effect on folding on its own, as it could still be holdase activity with this level of data.

After carefully reading the comments of Reviewer 1, it appears that the biochemical assays to describe the activity of flotillin have been misunderstood. Our intention is not to claim that flotillin shows foldase activity or that flotillin has a direct role in protein folding. Instead, we concur with the Reviewer's assessment that flotillin likely acts as a holdase. Flotillin holds unfolding proteins to stabilize them. Upon stabilization, these proteins may undergo correct folding but protein folding likely occurs without a direct intervention of flotillin. To ensure the accurate interpretation of our results, we modified the title, abstract and lines 357-359, 387-388, 476-480 and 506-508, to more clearly describe the potential holdase function of flotillin.

The experiment shown in Figure 5C was conducted with three independent experiments. The statistical analysis is shown in supplementary figure S8Fii. Panel 5C represents the LC-MS

profile of one of these biological replicates. It is not our intention to claim with this experiment that flotillin shows foldase activity. Rather, the purpose of this panel is to complement the previous panel Fig. 5A demonstrating flotillin holdase activity in stabilizing a fraction of unfolded proteins. The stabilized proteins may undergo correct folding but this occurs independently on flotillin activity. Correctly folded proteins shows enzymatic activity that can be detected by LC-MS, which is a good indicator of flotillin stabilizing activity but it should not be interpreted as flotillin playing a direct role in the refolding of these proteins.

We concur with the Reviewer's suggestion that protein solubility does not strictly equate to protein folding and have clarified this concept in lines 289-291 of this revised manuscript. We thoroughly revisited the entire text to clarify that our assay monitors protein solubility rather than protein folding. Nonetheless, protein insolubility provides us with a valuable assay to compare proteome stability across different backgrounds. Using this approach, we showed that a higher fraction of the proteome becomes unstable and precipitates in $\Delta floA$ compared to WT, indicating that flotillin plays a role in proteome stabilization. We used PBP2a as an example of a client protein that precipitates in the absence of flotillin to investigate the connection between protein insolubility and unfolding. *In vitro* (Fig. 5) and *in vivo* (Fig. 6) experiments showed that flotillin prevented PBP2a precipitation. The soluble fraction of PBP2a retained its enzymatic activity, whereas in the absence of flotillin, insoluble PBP2a became enzymatically inactive.

One specific thing for Fig 4A: show the total, soluble, and insoluble. At present, it's hard to tell where the proteins really go.

This revised manuscript includes a modified Fig. 4A showing total, soluble and insoluble protein fractions from DRM and DSM. The new panel shows similar protein content in DSM total and soluble fractions, whereas the DRM soluble fraction shows a significant drop in protein content compared to the total fraction. Consistently, the DRM insoluble fraction shows a much higher protein content than the DSM insoluble fraction. We would like to clarify that proteins from the insoluble fraction have been concentrated compared to the total and the soluble fractions, which are directly obtained from cell extracts. Thus, comparing protein abundance between fractions may lead to confusion.

In 4B- the conclusions are overstated here, there are other possibilities.

We have revised the conclusions extracted from Figure 4B according to the referee's suggestion (lines 300-307 of the revised version of the manuscript).

In 4C- since there's more in the insoluble fraction with the mutants, there should be less in the soluble fraction if they were loaded correctly, but this does not appear to be the case.

As previously addressed, establishing a direct comparison between the quantity of soluble and insoluble proteins may lead to confusion because the insoluble fraction has been concentrated while the soluble fraction has not. This has been clarified in the revised manuscript (lines 907-908) We have decided to show the insoluble proteins in this panel, given that a detailed characterization of soluble and insoluble proteins is shown in Fig. 4A.

In 5D- there's two issues. First, doing a 1:1 folded unfolded ratio could really not work as expected, since aggregation could continue. Second, the bands on the right of the gel are much

more faint than on the left, such that I don't think you would see any of the substrate bound by eye on the right side gel fractions anyway even if it was there.

The purpose of panel 5D was to support the results shown in Figure 5A-C. However, considering the reviewer's feedback, we realized that it may contribute to more confusion than clarity and have decided to remove it from the revised version of the manuscript.

However, showing a direct role in folding definitively is not required to make a worthy paper- the collection of results they have is suggestive of a role in proteostasis, which is interesting. The authors could remove results that have not been repeated and/or aren't statistically significant along these lines, tone down the wording on folding vs solubility, and could still have enough to publish here.

Our intention in this work is to show the critical contribution of flotillin to bacterial proteostasis through its holdase activity, by stabilizing unfolded proteins. We agree with this Reviewer's perspective that flotillin likely does not play a direct role in protein folding. This revised manuscript includes additional clarifications to avoid this confusion. Furthermore, we refined the interpretations of results and provided more detailed explanation of the methodology (e.g., folding vs. solubility). Main and supplemental figures have been reorganized and some figures/experiments have been removed (e.g., BLI or 1:1 co-incubation experiments) from the revised manuscript.

Minor points:

Fig 1A- a color problem, one of them should be gray.

It seems the gray color of $\Delta nfeD$ bar in panel 1A was inadvertently omitted during the conversion to low-resolution figures. We hope this revised version of the manuscript containing high-resolution figures will accurately show panel 1A with the correct colors.

For the experiments in Fig 1B and similar, couldn't find a materials and methods section, please add. For Fig 1D- can't find anything in the methods explaining how these were done as opposed to those in Fig 6.

The methods associated with Fig. 1B are detailed in the subsection titled Fluorescence Microscopy, in Material and Methods section (lines 1259-1266). Furthermore, the revised manuscript includes additional information (lines 132-38, 1497-1502 and 1511-1516) related to the methodology for animal infection experiments presented in panel 1D.

Fig 1C- typo in the iii label. Also, some parts aren't labeled.

We have corrected this accordingly in the revised manuscript.

Fig 1D- could you add a bit more summary sentences for the multipart panels? For this one in particular.

We have reformatted the figure legends and included a more detailed explanation for this panel in this revised manuscript.

Fig 2Di description in the text is quite unclear, not sure of the interpretation of the data by the authors here. The word remain is ambiguous.

We have rewritten this section (lines 183-187) to clarify the interpretation of data in Fig. 2Di.

For 3Ciii- couldn't find the materials and methods section for the immunodetection, please add.

The revised manuscript includes in the Material and Methods section (lines 1058-1063) additional information for the methodology used in the immunodetection assays.

4A: the total protein is total from the cell or total of membrane extracts?

This is the total protein of membrane extracts. We have clarified this point in lines 291-299 of the revised manuscript.

43i: unit issue on the y axis- looks like infinity grams/ml

This is probably caused by the conversion of the figure files to pdf. The revised manuscript includes double-checked figures converted with a different conversion system.

4E: there's a mislabel of ii in the caption as to the diagram.

We have corrected this in the revised manuscript.

Line 62: Pet peeve- nothing can be experimentally proven; this wording needs to be revised.

We have rewritten this sentence in the revised manuscript.

Reviewer #2 (Remarks to the Author):

In this manuscript, the authors provide compelling evidence for the critical role of flotillin membrane microdomains (FMM) in methicillin-resistant Staphylococcus aureus (MRSA) in confining and stabilizing unfolded proteins due to cellular stress. Using Cryo-EM analysis, they demonstrate that the PHB domain-containing proteins FloA and NfeD form clamp-shaped oligomers, creating a structure that holds accumulating unfolded proteins, thereby stabilizing them and aiding in their correct folding. Crucially, this process occurs without imposing a direct energy cost on the cell. The authors emphasize the significance of this mechanism for the survival of ATP-depleted bacteria and its impact on MRSA pathogenesis. The manuscript is well-written and illustrated. I have only a few minor concerns.

1. Figure 1A: The bars for bacterial survival of the $\Delta nfeD$ strain should be filled in gray.

The gray color of $\Delta nfeD$ bar in panel 1A likely went missing during the conversion to low-resolution figures. We hope that the revised version of the manuscript, which contains high-resolution figures, shows panel 1A with the correct colors.

2. Lines 122-123: The authors demonstrate that both H₂O₂-stressed cells and untreated cells collected at the stationary growth phase exhibit an increased number of foci compared to untreated, exponentially growing cells. Is this difference attributable to an up-regulation of the transcription and/or translation of FloA-NfeD? Furthermore, are there regulatory motifs identified in the upstream region of the operon?

Transcriptional upregulation of the operon in response to H₂O₂ is probably the most plausible explanation for the increased number of foci. We could not find any recognizable regulatory motif in the upstream region of the operon besides the housekeeping σ^A recognition motif. However, it is known that the stress-inducible σ^W upregulates the expression of the operon in the close relative bacterium *Bacillus subtilis* (Butcher and Helmann, 2006). σ^W is absent in *S. aureus* but an alternative mechanism may exist to upregulate the expression of the operon in *S. aureus* in response to different cellular stresses. Unfortunately, the recognition motifs of regulatory proteins in *S. aureus* are not as well characterized as they are in *B. subtilis*.

3. Lines 156-163: The authors propose that aromatic residues in the sterol-sensing domain (SSD)-like region (SSDL) are accountable for NfeD localization at FMM. However, MRSA cells do not have sterols in their cytosolic membranes. Which lipid species are responsible for NfeD recruitment at FMM? Has the binding of staphyloxanthin by NfeD been tested?

The aromatic residues within the transmembrane regions of NfeD enhance their hydrophobic properties, facilitating the recruitment and localization of NfeD at FMM. Although MRSA cells lack sterols in their membranes, they possess squalene-derived lipids with comparable physico-chemical properties to cholesterol, such as the carotenoid pigment staphyloxanthin. The localization of NfeD to FMM primarily stems from the hydrophobic affinity between FMM and the transmembrane segments of NfeD. This has been tested *in vivo*; a strain lacking staphyloxanthin also shows poor NfeD localization (Kricks, 2021). We clarified this in lines 154-156 of the revised manuscript.

4. Expanding on the aforementioned concern: Since NfeD specifically interacts and co-localizes with FloA (as demonstrated in Fig. 2 and detailed in lines 176-181), and FloA is known to localize in discrete foci across the cell membrane (as mentioned in lines 121-122 and depicted in Fig. 1B), and considering that this interaction is independent of the SSDL motif, I am intrigued by the uniform distribution of the NfeDSSDL-YFP construct (with an altered SSDL motif) in MRSA membranes, as observed in Fig. 2Bii. How could this be explained?

Understanding the specificity of interactions between flotillin and NfeD, or the client proteins, is a compelling aspect we intend to explore further as a continuation of the research presented in this manuscript. We hypothesize that recruiting proteins to FMM influences these interactions, potentially explaining why flotillin has evolved to be confined within raft-like membrane microdomains. Flotillin may not be highly selective in recognizing binding partners, as long as they are unfolded proteins. This is a versatile mechanism that ensures the surveillance of the entire proteome by flotillin, but at the same time requires the spatial compartmentalization of flotillin to FMM, to ensure that flotillin exclusively interacts with client proteins that need its assistance, and prevent unspecific interactions with other proteins that may cause metabolic interferences and cytotoxicity. This is probably why flotillin is typically confined in raft-like membrane microdomains across all kingdoms in life. Based on this, we hypothesize that protein recruitment to FMM is the selective step involved in flotillin-partner interaction. Thus, the capacity of NfeD to be recruited to FMM alongside flotillin may be a crucial factor driving its interaction with flotillin and NfeD variants whose localization is not restricted to FMM may show the interaction with flotillin affected.

5. Lines 274-277: the author mentioned "As FMM are highly hydrophobic membrane regions, they may provide a stabilizing environment to unfolded proteins that show their hydrophobic side-chains exposed to the solvent and contribute to isolating unfolded proteins from the rest of the cell to prevent unspecific interactions" Are FMM regions more hydrophobic than other

parts of the membrane? Sterols, terpenoids, and carotenoid lipid species contribute to membrane order and reduce fluidity, but as far as I understand, they do not increase hydrophobicity. The authors should provide references or further explanation to support this point.

The revised manuscript includes references (Sezgin et al., 2017 and Parasassi et al., 1997) and further information to support the hydrophobic properties of FMM (lines 282-284). FMM and other “raft-like” microdomains are packed and ordered membrane regions with reduced fluidity. Due to its tight molecular packing, water molecules are excluded, and their hydration level is low (i.e., more tightly packed membranes exclude water more efficiently). The higher hydrophobicity is a well-known property of raft-like membrane microdomains, and is used to design dyes for raft visualization using fluorescence microscopy, such as C-laurdan (Parasassi et al., 1997). As example, C-laurdan is a membrane dye that senses the level of membrane hydration in combination with fluorescence microscopy. The emission spectra of C-laurdan shifts depending on the aqueous content of the membrane. Based on this, raft-like membrane microdomains are identified from the rest of the cellular membrane based on their higher hydrophobicity and the different emission spectra of the C-laurdan. We used C-laurdan in Figure 4Fii and lines 344 of this manuscript.

Reviewer #3 (Remarks to the Author):

Review comments

Authors describes in this manuscript flotillin (FloA) and nfeD that constitute functional membrane microdomains (FMM) from Staphylococcus aureus (MRSA) and that using Cryo-EM studies, flotillins forms a clamp-shaped oligomer and also holds the accumulating unfolded protein, indicating that this FMM system contributes to MRSA viability during infection, due to these proteins stabilize unfolded proteins and favor their correct folding under ATP-independent pathway.

Many readers and I will be interested in these studies. The function of FMM scaffold protein flotillin have been unknown, but authors have first clarified their function.

Major points:

I think this article has large problem. In this manuscript, authors showed several cryo-EM studies on 1×FloA, 1×FloA-NfeD, 2×FloA-NfeD, 2×FloA-NfeD and PBP2a, but their resolution are ~9 Å, too low. In such a resolution, authors cannot describe precise molecular shapes. Authors emphasize the word “Cryo-EM” in the abstract, but I do not recommend it. In the abstract, authors should present other enormous studies such as Co-IP, CLMS, BLI, B2H, fluorescence microscopy and so on, rather than Cryo-EM. Even studies other than Cryo-EM are valuable for this manuscript. Or authors should emphasize that this cryo-EM studies are challenging to resolve small proteins (below 100 kDa) (which are described in Lines 204-208). This article presented much information about FloA, NdeD, and FMM to stabilize unfolded proteins. That is good, but I have some difficulties to read this manuscript, because too much Figures, Extended Figures, and supplements are present, and one figure contains complicated data. I want this manuscript more compact and readable to catch the importance as much as possible.

We followed the referee’s recommendations and the revised version of the manuscript includes several changes in the text to de-emphasize the use of Cryo-EM; specifically, we eliminated the reference to using cryo-EM from the abstract (line 32), added further caveats related to the low resolution of our particles (lines 221-223), the challenge to resolve small proteins (lines

212-216 and 221-223) and the need of additional work to obtain high-resolution structures to validate structural interpretations (lines 223-224). Furthermore, we highlighted the numerous molecular studies we conducted alongside cryoEM to corroborate our findings (lines 221-223).

The figures of the revised manuscript have been reorganized to compact the manuscript and make it more readable to a broader audience. Several panels have been removed (Supp S2D and E; Fig 5D and Eii) or moved to supplemental figures (Fig. 3Bii). We consider that this revised manuscript presents a more streamlined message to the reader.

Minor points:

Line 111: For the description “where is known to colocalize with flotillin (FloA)36,51”, ref 36 reported that FloT colocalize with FloA. Authors should explain NfeD and FloT in more precisely.

We have clarified this in lines 111-113 of the revised manuscript and included an additional reference addressing NfeD-flotillin interactions (Dempwolff et al., 2012 and Yokoyama et al., 2013).

Line 141: “one Nt transmembrane segment”, please spell out Nt at first.

We have corrected this accordingly in the revised manuscript.

Lines 195-196: “NfeD dimerization involves an OBL-OBL interaction”, please indicate why you can describe it using SEC analysis on Figure 2G, in which OBL shows one peak. I think this peak corresponds to a dimer. Is it correct ?

This is correct. The peak of OBL corresponds to a dimer. We clarified this in lines 202-204 of the revised manuscript and included changes in Fig. 2F and G to define the signals corresponding to monomeric or dimeric versions of NfeD.

Lines 243-244: This sentence are referred to Fig 3Biii and Extended Data Fig. 5D. Please explain more precisely, which peaks in Extended Data Fig. 5Dii correspond to the peptides in Diii. I cannot find data.

The supplemental Figure S5Dii of this revised manuscript now shows the experimental MS2 spectrum acquired after fragmentation, with the mayor product ions annotated in red. The peptide sequence identified is shown in the upper left corner of this panel.

Line 340: “hydrophobic interaction”, I do not understand why the recruitment of unfolded proteins at FMM is mainly driven by hydrophobic interaction. Please indicate.

We have clarified this concept in lines 282-286 of the revised manuscript. Folded proteins are usually soluble, showing the hydrophobic residues buried in the core, and the polar side chains are exposed to solvent. In contrast, unfolded proteins have exposed the hydrophobic side chains to the solvent; thus, they became insoluble and tend to precipitate in aqueous buffers (or the aqueous cytoplasm). We found that unfolded proteins accumulate at FMM attending to their hydrophobic affinity. FMM are highly hydrophobic regions that provide a stabilizing environment for unfolded proteins, and they isolate unfolded proteins from the rest of the cell to prevent unspecific interactions.

Extended data, Figure 3 Di: Arrowheads corresponding to FloA and NfeD show different points. Please correct.

We have corrected this accordingly in the revised manuscript.

Extended data, Figure 7 D: The legend i may correspond to ii, and the legend ii may correspond to to iii. There are no legends of i.

This issue has been corrected in the revised manuscript.

Extended data, Figure 8: Legends of D and E may be opposite.

We have corrected this accordingly in the revised manuscript.

REVIEWER COMMENTS

Reviewer #1 (Remarks to the Author):

Several of my comments have been addressed, but there are still two things to work out- the first is of medium importance (authors could cut the panel and conclusions and it would be ok), the second probably minor.

Fig 5C- I'm still not fully satisfied here. The quantification in Fig S8Fii doesn't match the eye test on Fig 5C assuming this one replicate shown is representative. At the 15 min mark, it looks like the amount of product is less than half what it is at 5 min, but the quantification makes it appear as though it is the same. This doesn't make sense. If the authors want to claim that folding is affected either directly or indirectly, this becomes important.

4C- in the rebuttal it says that there's a clarification in lines 907-908, but at that spot it's references, so it seems there's been a mistake here that needs clarification as to where I can find the clarification.

REVIEWER COMMENTS

Reviewer #1 (Remarks to the Author):

Several of my comments have been addressed, but there are still two things to work out- the first is of medium importance (authors could cut the panel and conclusions and it would be ok), the second probably minor.

Fig 5C- I'm still not fully satisfied here. The quantification in Fig S8Fii doesn't match the eye test on Fig 5C assuming this one replicate shown is representative. At the 15 min mark, it looks like the amount of product is less than half what it is at 5 min, but the quantification makes it appear as though it is the same. This doesn't make sense. If the authors want to claim that folding is affected either directly or indirectly, this becomes important.

Considering the reviewer's feedback, we realized this panel may contribute to more confusion than clarity and have decided to remove it from the revised version of the manuscript.

4C- in the rebuttal it says that there's a clarification in lines 907-908, but at that spot it's references, so it seems there's been a mistake here that needs clarification as to where I can find the clarification.

The clarification can now be found in lines 897-898 of the revised manuscript. These lines are part of the legend for Figure 4. Specifically, in the legend for panel 4A, we clarify that the insoluble protein fraction has been concentrated relative to the total and soluble fractions. This is to inform the readers that establishing a direct comparison between the quantities of soluble and insoluble proteins may be misleading.